# Comprehensive benchmarking of metagenomic binning tools reveals key factors for improved genome recovery

Jungyeon Kim[1], Nayeon Kim[1], Jun Hyung Cha[1], Junyeong Ma [1] & Insuk Lee [1,2] ✉

Metagenomic binning is essential for reconstructing prokaryotic genomes from metagenomic samples. We benchmarked various binning tools using Critical Assessment of Metagenome Interpretation (CAMI)-simulated, custom-simulated, and real metagenomic datasets, primarily focusing on short-read sequencing data. Our analysis highlights critical factors influencing binning efficacy: (i) Sequencing depth and taxonomic complexity strongly impact binning performance, while CAMI-simulated benchmarking datasets exhibit substantially lower complexity than human gut and environmental metagenomes, (ii) Chimeric genome rates vary widely across tools, (iii) Multi-sample binning is most effective with about 20 samples, as using too few or too many samples can reduce its benefits, and (iv) Binning efficacy was lower for single-end sequencing samples due to reduced contig quality and assembly fragmentation. Neural network-based tools consistently outperformed others in genome recovery from both real samples and simulated samples with realistic taxonomic complexity, though at higher computational cost. By integrating and refining genome bins from the top three binning tools, we recovered >30% more high-quality genomes than previous methods. This study provides practical guidance for improving metagenomic binning to facilitate the reconstruction of prokaryotic genomes.

The microbiome encompasses the collective genomes of microorganisms for a given environment and plays vital roles in ecosystems. Traditional culture-dependent microbiology was limited in studying microbial ecology[1]. To overcome this, sequencing environmental genomic fragments emerged as a powerful alternative. Whole metagenome shotgun sequencing is now widely used to analyze microbial genomes and genes without cultivation. A key advancement, genome-resolved metagenomics[2], reconstructs individual species genomes from metagenomic data, producing metagenome-assembled genomes (MAGs). This involves assembling short reads into contigs, clustering them via binning, and generating draft genomes, essential when cultured isolates are unavailable. Ensuring binned genome quality is critical for reliable downstream analysis.

Various approaches have been developed to improve the binning step in metagenomic analysis[3]. In the early stages, binning algorithms primarily relied on sequence composition features, such as k-mer frequency, based on the assumption that different genomes exhibit distinct sequence patterns, whereas sequences from the same genome share similar characteristics. While effective in many cases, this method faced challenges in distinguishing closely related genomes that possess similar genomic signatures[4,5]. To address these limitations, an alternative binning approach based on the relative abundance of reads in assembled contigs was introduced, operating on the principle that sequences from the same genome maintain consistent abundance levels within a sample and exhibit similar distribution patterns across multiple samples. More recently, hybrid binning

[1]Department of Biotechnology, College of Life Science and Biotechnology, Yonsei University, Seoul, Republic of Korea. [2]DECODE BIOME Co., Ltd, Incheon, Republic of Korea. ✉e-mail: insuklee@yonsei.ac.kr

**Fig. 1 | Overview of benchmarking programs and datasets. a** Metagenomic binners grouped into four categories based on how contig features (e.g., nucleotide composition and coverage) are processed and integrated. **b** Benchmarking datasets used in this study, including in-house simulation datasets, CAMI datasets and real microbiome datasets (gut, ocean, and soil). In-house simulations span multiple taxonomic complexity levels (number of species) and sequencing depths (dot size and colour, respectively; see keys in **b**). Created in BioRender. Yonsei, N. (2026) https://BioRender.com/mdhflt5. CAMI Critical Assessment of Metagenome Interpretation, NCBI National Center for Biotechnology Information, GI tract gastrointestinal tract.

strategies have emerged as the dominant approach, integrating both sequence composition and abundance-based methods[6]. By combining these two complementary metagenomic features, hybrid binning enhances accuracy and robustness, effectively overcoming the limitations of methods relying on either feature alone.

Nevertheless, integrating heterogeneous features and clustering them remains a significant challenge, leading to the development of numerous approaches. These algorithms can be broadly classified into four categories based on their machine learning model structures, especially how these features are processed and integrated prior to clustering (Fig. 1a and Supplementary Table 1).

The first category includes projection-based integration methods. CONCOCT[6] concatenate composition and abundance features, applying linear dimensionality reduction using principal component analysis, followed by clustering with a Gaussian Mixture Model. Similarly, binny[7] uses openTSNE for non-linear projection after concatenating the input features and performs clustering using the HDBSCAN algorithm.

The second category consists of tools based on probabilistic feature modeling. MaxBin2[8] multiplies the probability of two features before applying probabilistic modeling with the Expectation-Maximization algorithm. MetaBat[9] and MetaBat2[10] use weighted summation and geometric mean to integrate two features, refining distance measures to capture non-linear relationships. These integrated similarity scores are then used in a graph-based clustering framework with a modified label propagation algorithm. MetaDecoder[11] employs a modified Dirichlet Gaussian mixture model and a semi-supervised probabilistic approach to create pure clusters.

The third category includes tools that use neural networks to integrate composition and abundance features into latent representations, which are then used for clustering in the learned embedding space. VAMB[12] integrates oligonucleotide frequency and coverage features, using a variational autoencoder to generate latent representations, followed by iterative k-medoids clustering. SemiBin[13] introduces must-link and cannot-link constraints by splitting long contigs and incorporating taxonomic annotation, using a semi-supervised autoencoder for clustering. Its improved version, SemiBin2[14], refines constraint generation through random sampling and employs a self-supervised Siamese neural network for representation learning before clustering with DBSCAN algorithm. The

latest development, COMEBin[15], applies data augmentation to generate multiple perspectives for each contig, leveraging contrastive learning to obtain high-quality representations and performing clustering with the Leiden algorithm.

The last category includes MetaBinner[16], an ensemble-based integration method, where individual clustering results from different feature matrices are first generated and then integrated in a two-stage strategy guided by single-copy gene evaluation and scoring.

Historically, three distinct binning workflows—single-sample binning (independent binning of each sample on a single assembly), co-assembly binning (pooled assembly producing dataset-wide bins), and the multi-sample binning (leveraging multi-sample coverage but generating sample-specific bins)—were often misclassified or used interchangeably[12]. A prior binning study[17] primarily focused on the co-assembly workflow, considering it the default. However, this approach is now discouraged, as it may obscure strain-level and single-nucleotide variant information within individual samples. Given these challenges and the increasing sequencing depth, adopting both single-sample and multi-sample workflows is crucial for the thorough evaluation of metagenomic binning tools.

Previous evaluation of binning tools have relied on either Critical Assessment of Metagenome Interpretation (CAMI) simulation datasets[18,19] or real metagenomic datasets[20]. However, each source of evaluation data presents its own limitations. CAMI-simulated datasets facilitate accurate measurement of purity and completeness of MAGs under controlled sequencing depth and species complexity, but they may not capture all the constraints of microbial communities in real environments. Additionally, CAMI datasets typically exhibit much lower species complexity compared to actual microbiome samples. Real metagenomic datasets can better represent species complexity, yet they do not provide precise measures of MAG purity and completeness, nor do they permit systematic assessment of how sequencing depth and species complexity influence binning tool performance. Most crucially, a previous benchmarking study using real metagenomic datasets[20] have not considered genomic chimerism (i.e., the erroneous assignment of contigs from different sources to the same genome bin), which can vary significantly across different binning tools.

Our objective is to establish a comprehensive and reliable benchmarking workflow for metagenomic binning tools, with a focus

on short-read sequencing data that underpin most MAG cataloging projects. To account for the impact of sequencing depth and taxonomic complexity on binning performance, we generated custom simulation datasets with varying taxonomic complexity and sequencing depths, addressing the limitations of existing CAMI simulation datasets. For benchmarking with real metagenomic data, we also assessed genomic chimerism, which has been largely overlooked in previous evaluations. Our analysis revealed that the degree of genome chimerism varies widely across binning tools, representing a critical limitation of current methods. We also found that the number of samples used in multi-sample binning can strongly influence binning efficacy, and that binning performance is substantially lower when using single-end reads compared to paired-end reads, likely due to poorer contig quality and assembly fragmentation.

## Results

### Overview of metagenomic binning tool benchmarking study

In this study, we evaluated nine genome binning tools, including widely used and recently developed neural network-based methods: CONCOCT, MaxBin2, MetaBAT2, VAMB, MetaDecoder, Binny, SemiBin2, MetaBinner, and COMEBin (Fig. 1a and Supplementary Table 1). Most programs were tested using their default settings; however, single-sample workflows were applied even for tools originally designed for co-assembly to preserve the unique characteristics of each sample. For tools explicitly supporting sample-specific multi-sample binning, such as SemiBin2 and VAMB, we also evaluated their multi-split modes (SemiBin2-multi and VAMB-multi). Additionally, we assessed three binning refinement tools: MetaWRAP refinement module[21], DAS Tool[22], and MAGScoT[23].

We used three types of datasets for benchmarking: CAMI simulation datasets, custom simulated metagenomic datasets, and real metagenomic datasets (Fig. 1b). Various CAMI datasets have been used to evaluate binning tools[18], but most only provide ground truth mappings for pooled assemblies (i.e., mapping of gold-standard contigs to reference genomes) rather than for individual samples. To address this limitation, we selected the toy human dataset from the second CAMI challenge, which includes single-sample gold-standard mappings for individual samples. We refer to this dataset as the CAMI2 human dataset, which consists of 10 samples each from three sites—GI tract, oral, and airway—with an average sequencing depth of approximately 5 Gb per sample.

However, the CAMI2 human dataset exhibits relatively low taxonomic complexity, with the number of reference genomes per sample ranging from 35 to 274. The median values are 54.3 for the GI tract dataset, 178.8 for oral, and 164.6 for airway, with an overall mean of 132.6. This limited complexity reduces its relevance to real metagenomic samples, which typically contain a far greater diversity of taxa. To address this, we generated custom simulated datasets using CAMISIM[24], incorporating de novo community design based on NCBI reference bacterial genomes. These datasets include four levels of sequencing depth (2.4, 5, 7.2, and 10 Gb) and five levels of taxonomic complexity (60, 150, 600, 1000, and 1500 species), resulting in a total of 20 simulated metagenomic datasets. Each dataset is labeled as c[taxonomic complexity]_sequencing depth (e.g., c60_2.4 Gb represents a dataset with 60 species genomes and a sequencing depth of 2.4 Gb). Consistent with the CAMI2 human dataset, each simulated dataset consists of 10 samples, while also considering the computational time required for dataset generation.

We also utilized real metagenomic samples from human gastrointestinal, soil, and ocean microbiomes (Supplementary Table 2). For each environment, 20 publicly available metagenome sequencing samples were randomly selected from studies used to construct recent genome catalogs, ensuring that the samples passed quality control criteria (Supplementary Data 1). Additionally, we collected human gut

datasets with varying sequencing depths and read formats (single-end or paired-end sequencing).

### Assessment of metagenomic binning tools on CAMI datasets with low taxonomic complexity

Quality assessment for both the CAMI and custom simulation datasets was conducted using the AMBER[25] package, which provides various binning quality metrics, including completeness, purity, and adjusted rand index (ARI). Completeness measures the proportion of sequences successfully assigned to bins, while purity measures the proportion of correctly assigned sequences within each bin. ARI, a widely used clustering evaluation metric, assesses how well predicted bins correspond to reference genomes.

We first evaluated binning performance using three CAMI2 human datasets: Airway, GI tract, and Oral. Among the binning tools, binny, VAMB-multi, and MetaBinner exhibited the highest purity, while CONCOCT, MaxBin2, and COMEBin showed the lowest purity (Fig. 2a). MetaBinner and binny also achieved the highest ARI scores, whereas MetaBAT2, VAMB, and CONCOCT exhibited particularly low ARI values. Despite producing high-purity bins, MetaBAT2 and VAMB still showed low ARI scores, suggesting that these tools generate highly pure bins but struggle to comprehensively reconstruct genomes. Notably, completeness scores were lower across binning tools for the Airway and Oral datasets compared to the GI tract dataset. Given that the taxonomic complexity of the Airway and Oral datasets (averaging 165 and 179 species per sample, respectively) is much higher than that of the GI tract dataset (averaging 54 species per sample) (Supplementary Table 3), this suggests that the completeness metric is strongly influenced by the taxonomic complexity of a given metagenomic sample.

For real de novo genome assembly projects, the primary interest lies in how many genomes meeting specific quality criteria can be recovered, rather than focusing solely on the average completeness or purity of genome bins. Therefore, we assessed metagenomic binning tools using quantity metrics, which count the number of genome bins passing minimum purity and completeness thresholds (Fig. 2b). Based on the number of genome bins with purity ≥ 95% across varying completeness levels, binny, VAMB-multi, and MetaBinner showed the highest performance. Notably, additional tools such as COMEBin, MetaDecoder, SemiBin2, and SemiBin2-multi also demonstrated strong performance when assessed using these quantity metrics.

Finally, we summarized benchmarking results by integrating both quality and quantity metrics (Fig. 2c). Neural network-based tools, including COMEBin, SemiBin2, SemiBin2-multi, and VAMB-multi, consistently outperformed other tools, while previously widely used tools like CONCOCT, MaxBin2, and MetaBAT2 repeatedly underperformed, highlighting their limited ability to recover genomes even from low-complexity metagenomic samples.

### Evaluation of metagenomic binning tools on custom simulated datasets with varying taxonomic complexity and sequencing depth

While the taxonomic complexity of most CAMI samples remains under 200 genomes, real microbial communities often exceed several hundred to over a thousand. To ensure that benchmarking results are applicable to projects involving real samples, it is crucial to conduct evaluations using simulation datasets with higher taxonomic complexity. Therefore, we generated 20 custom simulation datasets using CAMISIM, incorporating a range of taxonomic complexities (c_60, c_150, c_600, c_1000, c_1500) and sequencing depths (2.4, 5, 7.2, and 10 Gb), each representing different combinations of taxonomic complexity and sequencing depth (Supplementary Fig. 1a, b).

Through evaluation using our custom simulated datasets, we demonstrated the impact of sequencing depth and taxonomic complexity on metagenomic binning quality metrics. Most notably,

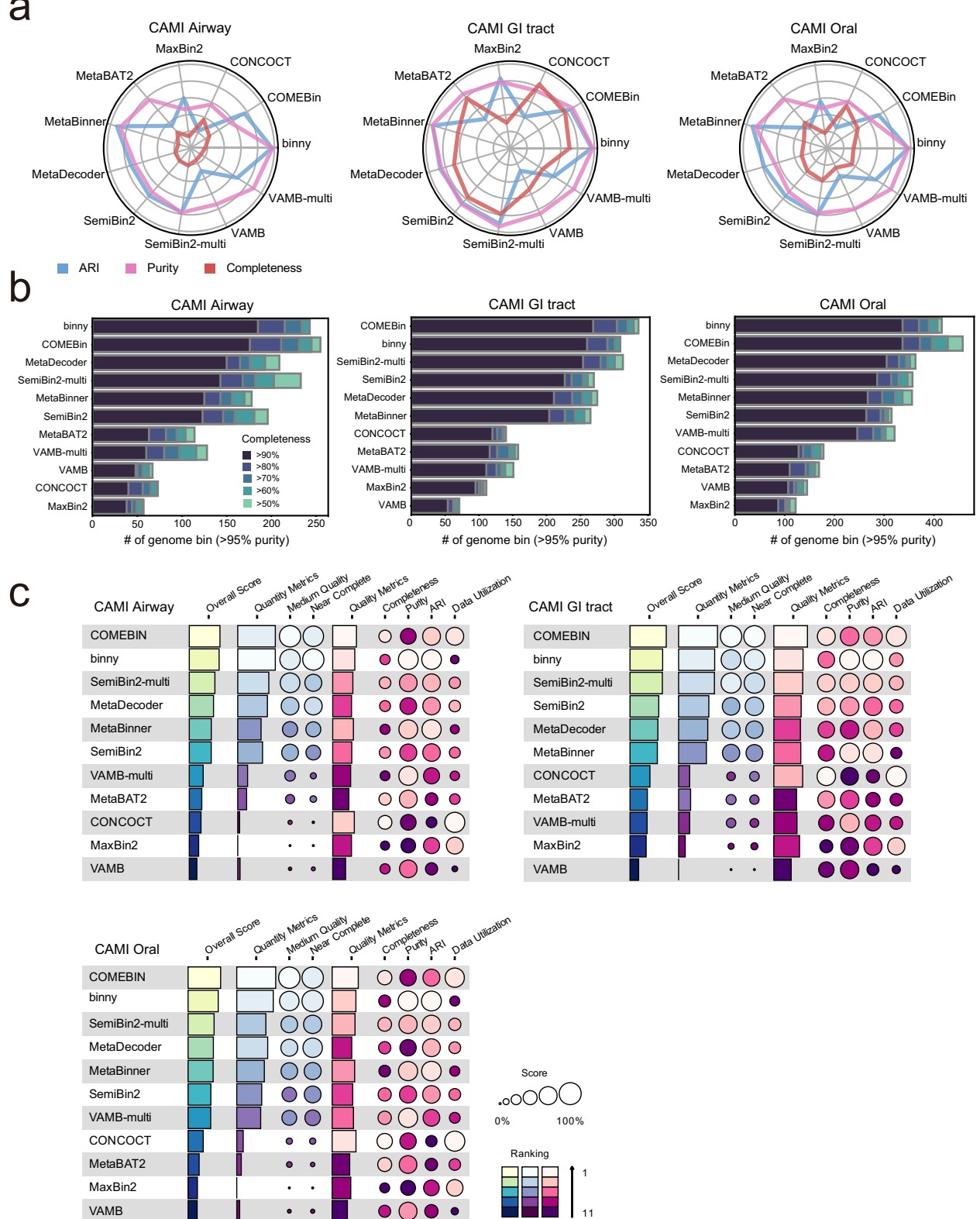

**Fig. 2 | Metagenomic binning benchmarks on CAMI human toy microbiome datasets. a** Radar plots summarizing binning performance across CAMI2 body sites (Airway, GI tract, and Oral) using ARI (blue), purity (magenta) and completeness (red). **b** Number of bins with contamination <5%, categorized by completeness thresholds of 50, 60, 70, 80, and 90% for each body site in CAMI2 human dataset. **c** Summary plot of the benchmarking results for CAMI2 (circle size indicates score; colour indicates ranking; see "Methods" for construction and scoring). ARI adjusted rand index, GI tract gastrointestinal tract.

completeness scores positively correlated with sequencing depth (Spearman correlation $\rho = 0.20$, $p < 7.6e\text{-}44$), while showing an inverse correlation with taxonomic complexity (Spearman correlation $\rho = -0.55$, $p < 1e\text{-}100$) across all binning tools (Fig. 3a and Supplementary Fig. 1c). This suggests that the lower binning efficacy observed in samples with high taxonomic complexity can be partially mitigated by deeper sequencing. In contrast, purity and ARI scores remained relatively stable across varying sequencing depths and taxonomic complexities, indicating that these quality metrics are more reflective of the intrinsic characteristics of each binning tool. However, previously widely used tools such as CONCOCT, MaxBin2, and MetaBAT2 showed a noticeable decline in purity and ARI at taxonomic complexities of 600 species or higher, highlighting their limited efficacy for complex microbiomes, such as the human gut microbiome. Notably, VAMB consistently showed low ARI scores across all datasets, indicating its inherent limitations in metagenomic binning. However, the performance of VAMB improved substantially when applied in multi-sample binning mode, suggesting that VAMB should be used with multi-sample binning for optimal performance in metagenomic binning projects.

Next, we evaluated metagenomic binning tools using a quantity metric—the number of near-complete (NC) genome bins (completeness ≥ 90%, purity ≥ 95%). Across all tools and taxonomic complexities, the number of NC genome bins increased with sequencing depth. Notably, the performance gap between the top three tools and the bottom three tools widened substantially in samples with high taxonomic complexity, particularly at sequencing depths of 7.2 Gb or higher (Supplementary Fig. 2a). Including SemiBin2, the fourth-best tool, all top-performing tools that showed a strong positive response to sequencing depth in datasets with more than 600 species were neural network-based (Supplementary Fig. 2b). This highlights that the challenges posed by high taxonomic complexity can be effectively mitigated when sufficient sequencing depth is coupled with neural network-based binning tools.

CONCOCT, which performed well at low taxonomic complexity (60 species), became one of the worst tools at complexities of 600 or higher, indicating its sensitivity to community complexity. MaxBin2 showed a similar trend, performing reasonably well at low complexity but deteriorating sharply in more complex samples (Supplementary Fig. 2b), suggesting that probabilistic-based binning approaches are more vulnerable to increasing complexity.

COMEBin, SemiBin2-multi, and VAMB-multi consistently ranked among the top three tools, while VAMB, MaxBin2, CONCOCT, and MetaBinner were consistently among the worst at taxonomic complexity levels of 150 or higher. Summary comparisons of both quantity and quality metrics (Fig. 3b) reinforced these findings, highlighting COMEBin, VAMB-multi, and SemiBin2-multi as the best performers, and VAMB, MaxBin2, and MetaBinner as the weakest. Notably, all three top performers leverage neural network models, indicating their suitability for high-complexity, high-depth metagenomic datasets. Additionally, Binny, which employs iterative refinement, also demonstrated promising performance.

### Benchmarking metagenomic binning tools on real metagenomic datasets, including chimeric genome evaluation

Although we simulate metagenomes with varying sequencing depths and taxonomic complexities, synthetic data may not fully capture the constraints of real microbial communities. To address this, we compiled metagenomic datasets from human gut, ocean, and soil samples with diverse sequencing depths and assessed binning performance. As ground truth genome assignments are unavailable for these real datasets, we evaluated genome bin completeness and contamination using CheckM2, which estimates these metrics through a machine-learning model trained on genome-wide amino-acid composition and gene-content features[26]. For genome bins with contamination <5%, COMEBin consistently recover more genomes than other tools in human gut and soil samples (Fig. 4a). Consistent with the simulated data results, neural network-based tools generally showed superior genome recovery, while previously widely used tools such as CONCOCT, MaxBin2, and MetaBAT2 performed poorly in gut and ocean samples. In soil samples with similar sequencing depths, most binning tools struggled to recover high-quality genomes. Given the exceptionally high microbial diversity in soil[27], the extreme taxonomic heterogeneity presents a significant challenge for metagenomic binning. Notably, COMEBin was the only tool capable of effectively reconstructing genome bins even in these complex soil metagenomes.

For genome binning using real metagenomic data, it is essential to consider an additional quality aspect: genomic chimerism. Genome contamination can be categorized into two types: redundant and non-redundant contamination. Redundant contamination arises from closely related lineages and results in surplus genomic fragments, while non-redundant contamination introduces foreign fragments, producing chimeric genomes. Importantly, non-redundant contamination is difficult to detect by CheckM2. To identify chimeric genomes, we applied Genome UNCluttere (GUNC)[28], which filters genome bins with a clade separation score (CSS) > 0.45, capturing chimerism involving fragments from different genera or higher taxonomic ranks. Notably, filtering for chimeric genomes reduced the number of NC genome bins (completeness ≥ 90%, contamination <5%) to varying degrees across tools (Fig. 4b and Supplementary Fig. 3). For example, MaxBin2 and MetaBinner exhibited the highest proportions of chimeric genomes, while genome bins recovered by VAMB and VAMB-multi showed much lower chimeric genome rates. The frequency of chimeric genomes did not correlate with sequencing depth, highlighting that genome chimerism is an intrinsic factor dependent on the binning algorithm itself. These findings suggest that potential genome chimerism should be taken into consideration when evaluating metagenomic binning tools, although the absolute values of contamination rates should be interpreted with caution.

We combined quality and quantity metrics to evaluate the overall performance of each binning tool on the real metagenomic dataset Gut 7.2 Gb. The top-ranked tools were all neural network-based, including COMEBin, SemiBin2, SemiBin2-multi, and VAMB-multi (Fig. 4c). We further summarized the overall scores across all simulated and real datasets used in this study, consistently identifying these four neural network-based tools as the top performers (Fig. 4d). Notably, COMEBin achieved the highest overall performance in both simulated and real datasets. These results reinforce that neural network-based binning tools represent the current best practice for accurate and efficient recovery of genomes from metagenomic samples.

Improving genome recovery often relies on computationally intensive algorithms that incur high computational costs, posing challenges for large-scale metagenomic projects. To assess computational efficiency, we benchmarked tools on the Gut 7.2 Gb dataset using a 5-core CPU configuration ("Methods"). COMEBin and binny showed markedly longer runtimes than other tools (Fig. 4e and Supplementary Table 4), likely reflecting COMEBin's data augmentation and binny's iterative processing. With GPU-based computing now widely accessible, a GPU-accelerated COMEBin achieves comparable performance to other neural network-based binners (~1 h per sample) and enables practical large-scale analyses (e.g., 1000 samples in ~4 days using 10 GPUs). Although its runtime remains longer than MetaBAT2 (5 min), COMEBin's recovery of more than twice as many genomes (505 vs. 204) justifies the computational investment. Once established, such reference genome catalogs support downstream analyses for years, further validating the value of GPU resources.

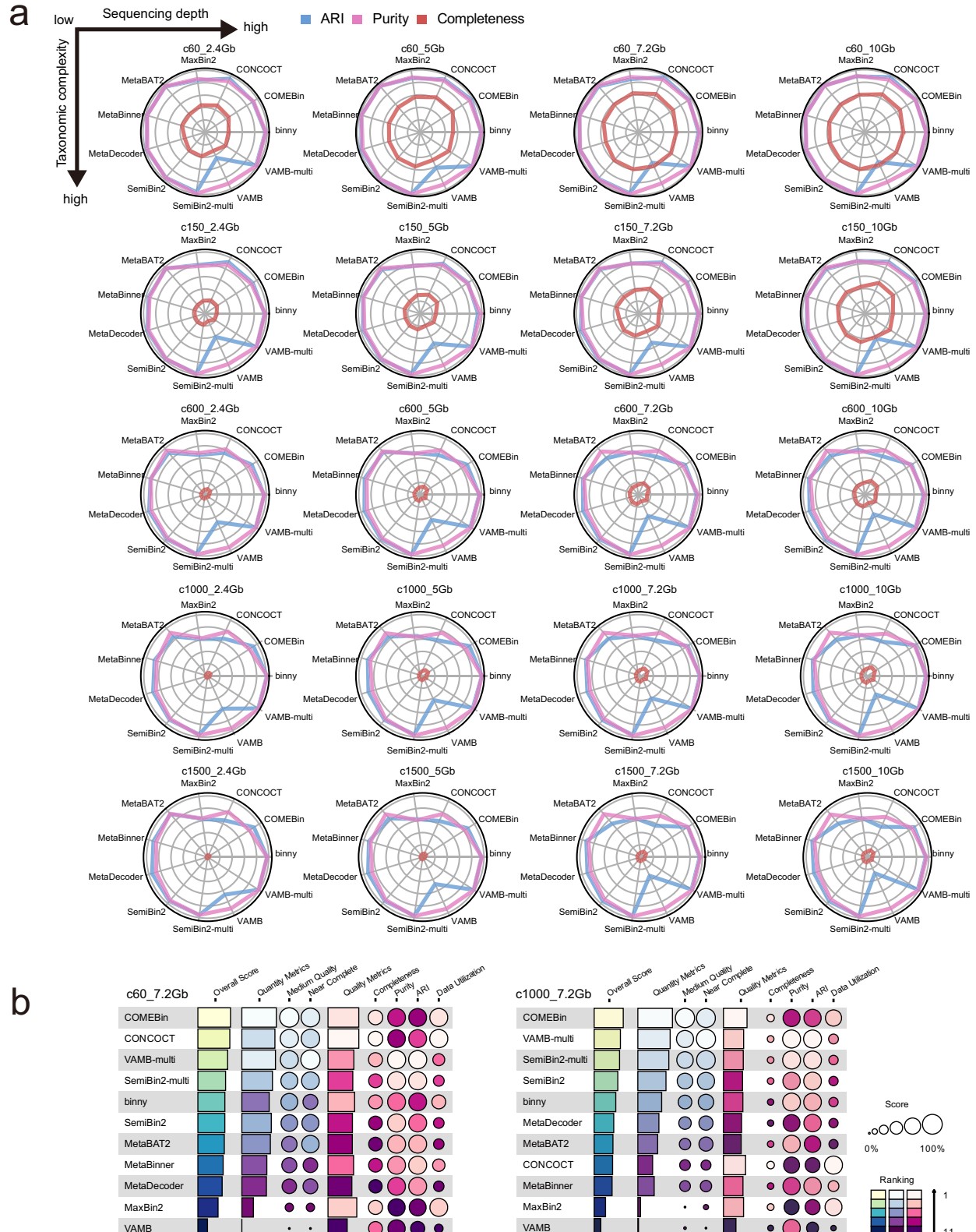

**Fig. 3 | Metagenomic binning benchmarks on In-house simulation datasets.**
**a** Radar plots of ARI (blue), purity (magenta), and completeness (red) across four
community complexity levels and five sequencing depths (arranged from low
complexity/shallow depth to high complexity/deep depth; see arrows in (**a**)).

**b** Summary plot of the benchmarking results for representative in-house simula-
tions (c60_7.2 Gb and c1000_7.2 Gb; circle size indicates score; colour indicates
ranking; see "Methods" for construction and scoring).

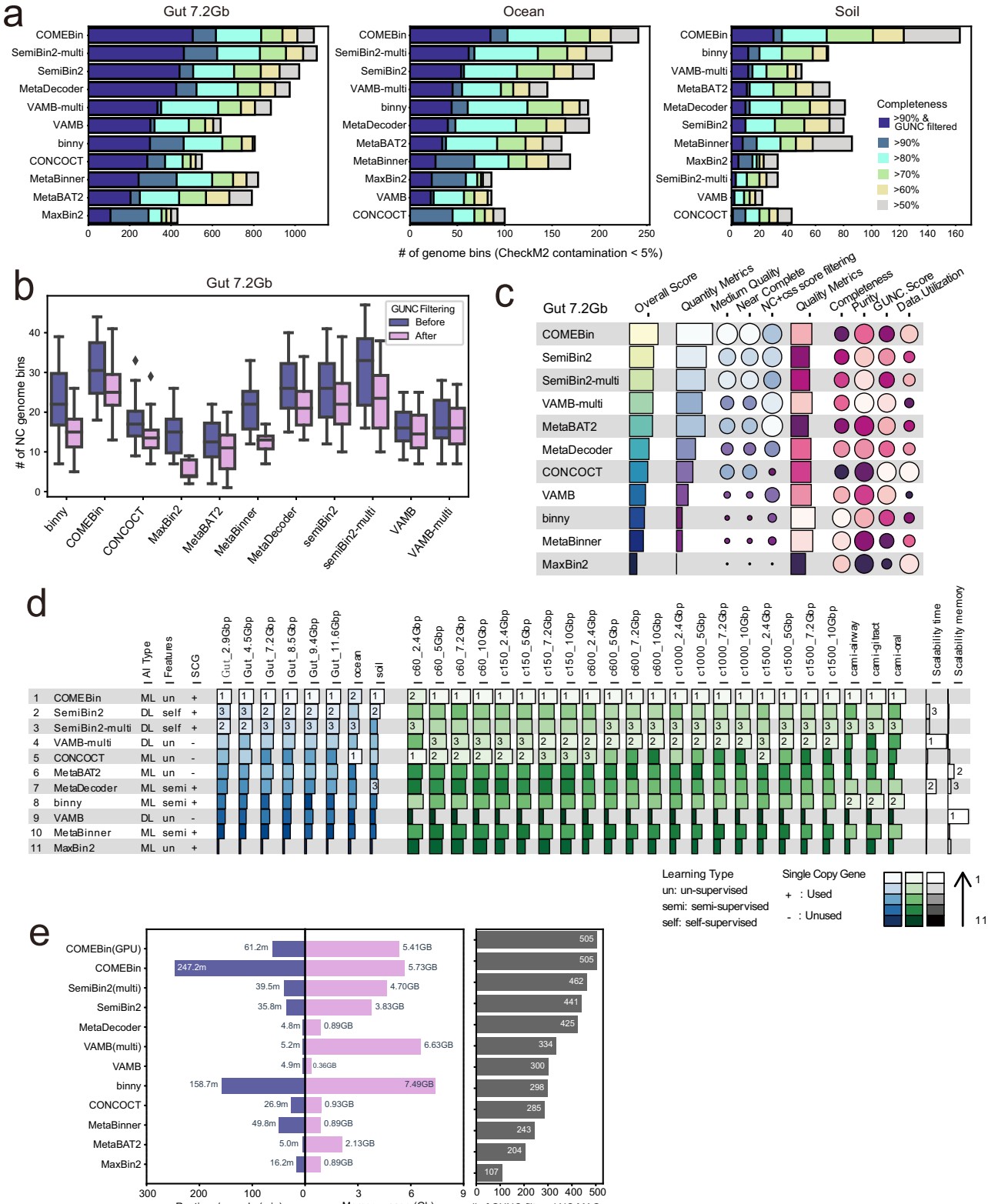

**Multi-sample binning improves genome recovery, but using too few or too many samples can diminish its benefits**

Multi-sample binning has been shown to improve genome recovery compared to single-sample binning, albeit with increased computational cost and resource demands[29]. However, prior evaluations were limited to MetaBAT2 applied to rumen microbiome samples. More recently, a study constructing a vaginal microbial genome catalog demonstrated that multi-sample binning with MetaBAT2 significantly

increased genome recovery, although the benefit plateaued after approximately 20 samples[30]. Building on these observations, we assessed how the number of samples affects multi-sample binning performance for VAMB and SemiBin2 using a Gut 7.2 Gb dataset. We evaluated genome recovery across increasing subsets of randomly selected 5, 10, 20, 30, and 40 samples. Notably, both VAMB-multi and SemiBin2-multi showed enhanced genome recovery with multi-sample binning compared to single-sample binning for human gut

**Fig. 4 | Metagenomic binning benchmarks on real microbiome datasets.**
**a** Benchmarking results across gut_7.2 Gb, ocean, and soil datasets, shown as numbers of genome bins with <5% contamination stratified by completeness thresholds (50–90%) and with additional GUNC filtering (CSS score <0.45; completeness categories and GUNC-filtered bins are shown in the key in (**a**)). **b** Near-complete (NC) MAG counts before and after GUNC filtering for each binning tool in gut_7.2 Gb ($n = 20$ samples per tool; colours indicate filtering status: Before, blue; After, pink). Box lengths represent the interquartile range (IQR) of the data, and whiskers extend to the lowest and highest values within $1.5 \times$ IQR from the first and third quartiles, respectively. The center line indicates the median, and all outliers

are shown. **c** Summary plot of the benchmarking results for the gut_7.2 Gb dataset. **d** Final ranking of binning tools, calculated by assigning equal weight to all datasets and ranking tools accordingly. The results are sorted in descending order based on final ranking, providing an overall performance comparison across datasets. In (**c** and **d**), colour indicates ranking and marker size indicates score (see keys in the panels). **e** Computational resource usage (runtime per sample and peak memory) measured under identical conditions (five CPU cores or one GPU) together with the number of GUNC-filtered NC MAGs; GPU-based and multi-sample modes are indicated in the labels. NC Near Complete, GUNC Genome UNClutterer, ML Machine learning, DL Deep learning, SCG single-copy marker gene.

---

metagenome samples, but only when using more than 10 samples. Using over 20 samples did not offer additional benefits and could even diminish the benefit for SemiBin2-multi (Fig. 5a). Additionally, an optimal average quality score of genome bins was observed around 20 samples for both binners (Fig. 5b). Repeating the evaluation with the Gut 4.5 Gb dataset produced consistent trends (Fig. 5c, d). These findings suggest that while multi-sample binning can improve genome recovery, using too few or too many samples may impair its effectiveness.

Based on these findings, we recommend using approximately 20 samples for multi-sample binning. Using significantly fewer samples tends to decrease effectiveness, while using substantially more can waste computational resources and potentially degrade binning performance. This consistent pattern across tools like MetaBAT2, VAMB, and SemiBin2 suggests that the optimal sample count for multi-sample binning could be generalized to other binning tools. Consequently, we advise against using overly small or excessive numbers of samples in multi-sample binning.

## Binning efficacy is lower for single-end sequencing samples due to poorer contig quality

A considerable number of metagenome sequencing datasets deposited in public databases were generated using single-end sequencing. In the early years of metagenomic research, single-end sequencing was commonly employed for microbial profiling due to its lower cost compared to paired-end sequencing. To assess the impact of single-end sequencing on genome recovery using metagenomic binning tools, we compiled 40 human gut metagenome datasets generated using single-end sequencing, with sequencing depths of approximately 7 and 10 Gb (20 samples for each depth) (Supplementary Data 1). Notably, single-end sequencing samples consistently produced suboptimal results compared to paired-end sequencing samples, yielding significantly fewer NC genome bins per sample (Fig. 6a). We hypothesize that the reduced binning efficacy observed for single-end sequencing samples may be attributed to lower contig quality and increased assembly fragmentation, both of which likely result from the inherent limitations of single-end sequencing data.

To test this hypothesis, we generated derived single-end sequencing samples by taking only the forward reads from paired-end sequencing samples in the Gut 7.2 Gb dataset. These derived single-end sequencing samples, which were created by subsampling reads from each paired-end sequencing sample, also resulted in reduced sequencing depth, further affecting binning efficacy. To minimize the impact of sequencing depth, we also created half paired-end sequencing samples by taking 50% of the paired-end sequencing reads. When we performed contig assembly and binning processes on the derived single-end sequencing samples, the number of NC genome bins was substantially reduced compared to both the original paired-end and half paired-end sequencing samples (Fig. 6a), confirming the results observed with real datasets. To investigate whether the reduced binning efficacy of derived single-end sequencing samples is due to poorer-quality assembled contigs, we compared the average contig count and contig length for the different sample types. As

anticipated, we observed substantially lower contig count and contig length for the derived single-end samples compared to both the original paired-end and half paired-end sequencing samples (Fig. 6b, c). These results suggest that the structural advantage of paired-end sequencing reads, which provide spatial linkage information, is a key factor in enhancing both assembly and binning performance, regardless of sequencing depth.

We next evaluated the performance of the same binning tools on single-end sequencing data using 40 human gut metagenome samples with sequencing depths of approximately 3 and 7 Gb (20 samples per depth; Supplementary Data 1). Notably, COMEBin, SemiBin2-multi, and VAMB-multi consistently achieved top-tier genome recovery in single-end datasets, mirroring their performance with paired-end sequencing data (Fig. 6d). Overall performance metrics showed similar trends, with CONCOCT and single-sample SemiBin2 also exhibiting competitive results (Fig. 6e). These findings indicate that COMEBin, SemiBin2-multi, and VAMB-multi represent robust choices for metagenomic binning across both single- and paired-end sequencing datasets at varying sequencing depths.

## Ensemble of complementary binning tools substantially enhances genome recovery

Different metagenomic binning tools may exhibit varying sensitivity to different genomes, meaning that integrating complementary results from multiple tools could enhance genome recovery from metagenomic samples. To investigate this complementarity, we clustered the NC genome bins obtained from each tool using the dereplication tool dRep[31], applying a 95% average nucleotide identity (ANI) threshold and an 80% coverage cutoff. This clustering grouped similar genomes into species clusters, allowing us to identify both shared and tool-specific species with recovered genomes. Although most NC genome bins were recovered by multiple binning tools, particularly including all neural network-based tools, a substantial portion of NC genome bins were uniquely recovered by COMEBin, SemiBin2-multi, and VAMB-multi with no overlap among them (Fig. 7a). The ability to recover unique species is critical for expanding taxonomic diversity of microbial genome catalogs, especially in environments harboring rare or highly diverse taxa. These results suggest that these three tools not only complement other tools but also complement each other. Therefore, we selected these three neural network-based binning tools for further investigation to assess whether an ensemble approach could recover substantially more genomes.

To evaluate the effectiveness of bin refinement, we compared a commonly used MetaWRAP refinement module and DAS Tool with a more recently developed tool, MAGScoT, using real human gut metagenome datasets with varying sequencing depths (Gut 2.9 and 8.5 Gb). Across both datasets, MetaWRAP consistently outperformed DAS Tool and MAGScoT in the recovery of GUNC-filtered NC genome bins (Fig. 7b). Notably, while DAS Tool and MAGScoT retrieved comparable number of NC genomes before GUNC filtration, they produced significantly more chimeric genomes, ultimately resulting in fewer genome bins than MetaWRAP after GUNC filtration. We further evaluated computational efficiency of bin refinement tools using a 5-core

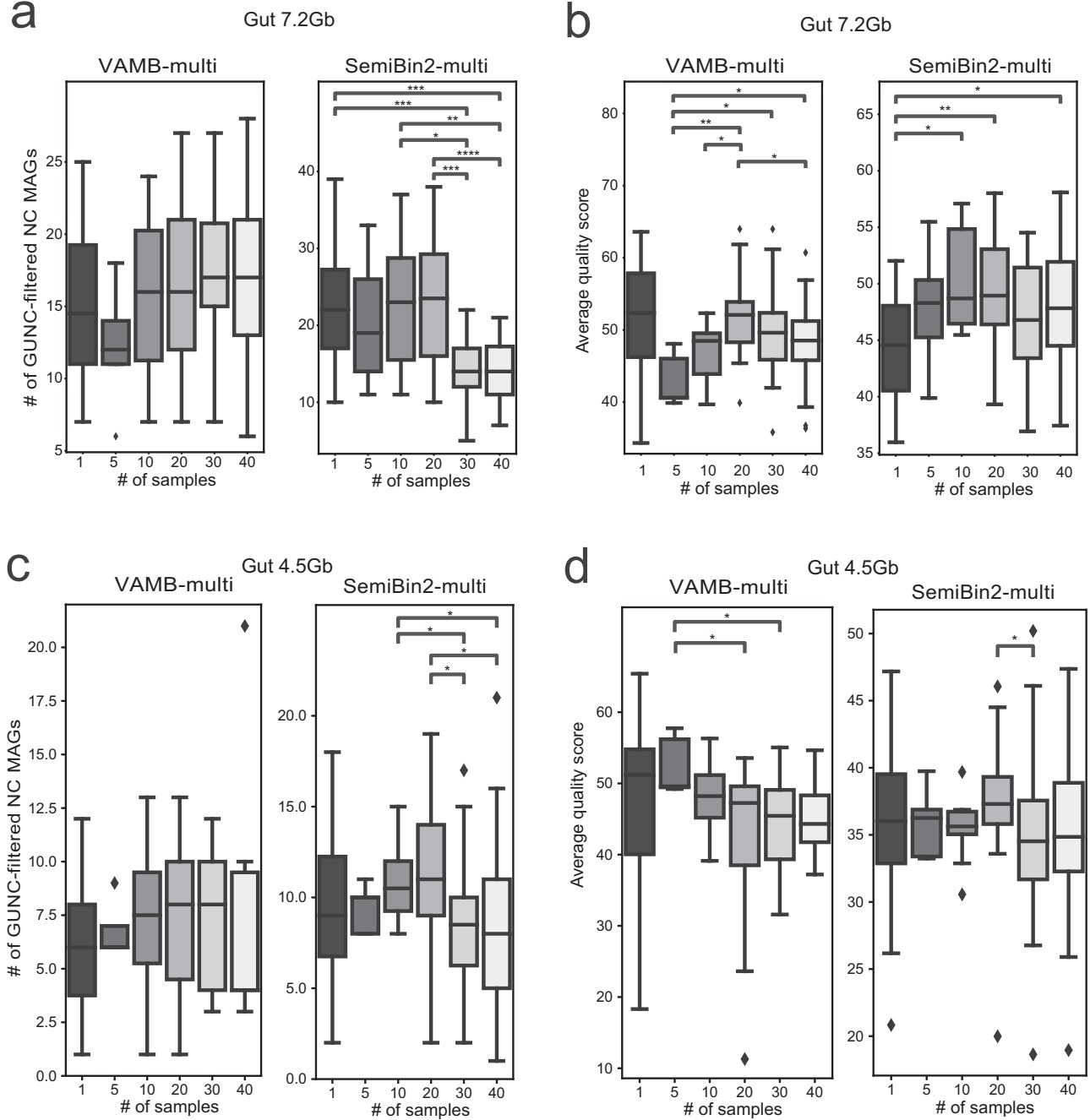

**Fig. 5 | Investigation of impact of the number of samples on multi-sample binning performance. a**, **b** Number of GUNC-filtered near-complete (NC) MAGs and average quality scores reconstructed by VAMB and SemiBin2 using the multi-sample mode in the gut_7.2 Gb dataset, measured across increasing numbers of input samples. **c**, **d** Corresponding results for the gut_4.5 Gb dataset under identical conditions. The x-axis indicates the number of samples included per run; box plots summarize sample-level values (NC MAG counts for (**a**, **c**); quality scores for (**b**, **d**)). The 1-sample setting comprises 20 independent 1-sample runs using different samples ($n = 20$); for all other settings, $n$ equals the number of samples included in the run (as indicated on the x-axis). Statistical significance was assessed using the two-sided Mann–Whitney $U$ test (stars indicate unadjusted $P$ values; exact $P$ values are provided in the Source Data, with Benjamini–Hochberg-adjusted $P$ values reported for reference; *$p \leq 0.05$, **$p \leq 0.01$, ***$p \leq 0.001$). Box lengths represent the interquartile range (IQR) of the data, and whiskers extend to the lowest and highest values within $1.5 \times$ IQR from the first and third quartiles, respectively. The center line indicates the median, and all outliers are shown.

CPU configuration and found that MetaWRAP required considerably longer runtime (~85 min per sample) than DAS Tool (~8 min per sample) and MAGScoT (~2.5 min per sample) (Supplementary Table 5). Nevertheless, considering that a well-curated genome catalog can serve numerous downstream analyses for years, we believe that this computational investment can be justified when sufficient CPU resources are available (e.g., refinement of 1000 samples can be completed in 3 days with 100 CPU cores). Focusing primarily on

maximizing genome recovery, we propose an optimal ensemble binning pipeline that combines COMEBin, SemiBin2-multi, and VAMB-multi, followed by bin refinement using MetaWRAP. We refer to this pipeline as MetaWRAP (COMEBin, SemiBin2-multi, VAMB-multi).

We further compared this proposed pipeline to ensemble binning pipelines used in the development of highly cited genomic catalogs, such as DAS Tool with CONCOCT, MaxBin2, and MetaBAT2 used for the UHGG[32] catalog, and MetaWRAP with CONCOCT, MaxBin2, and

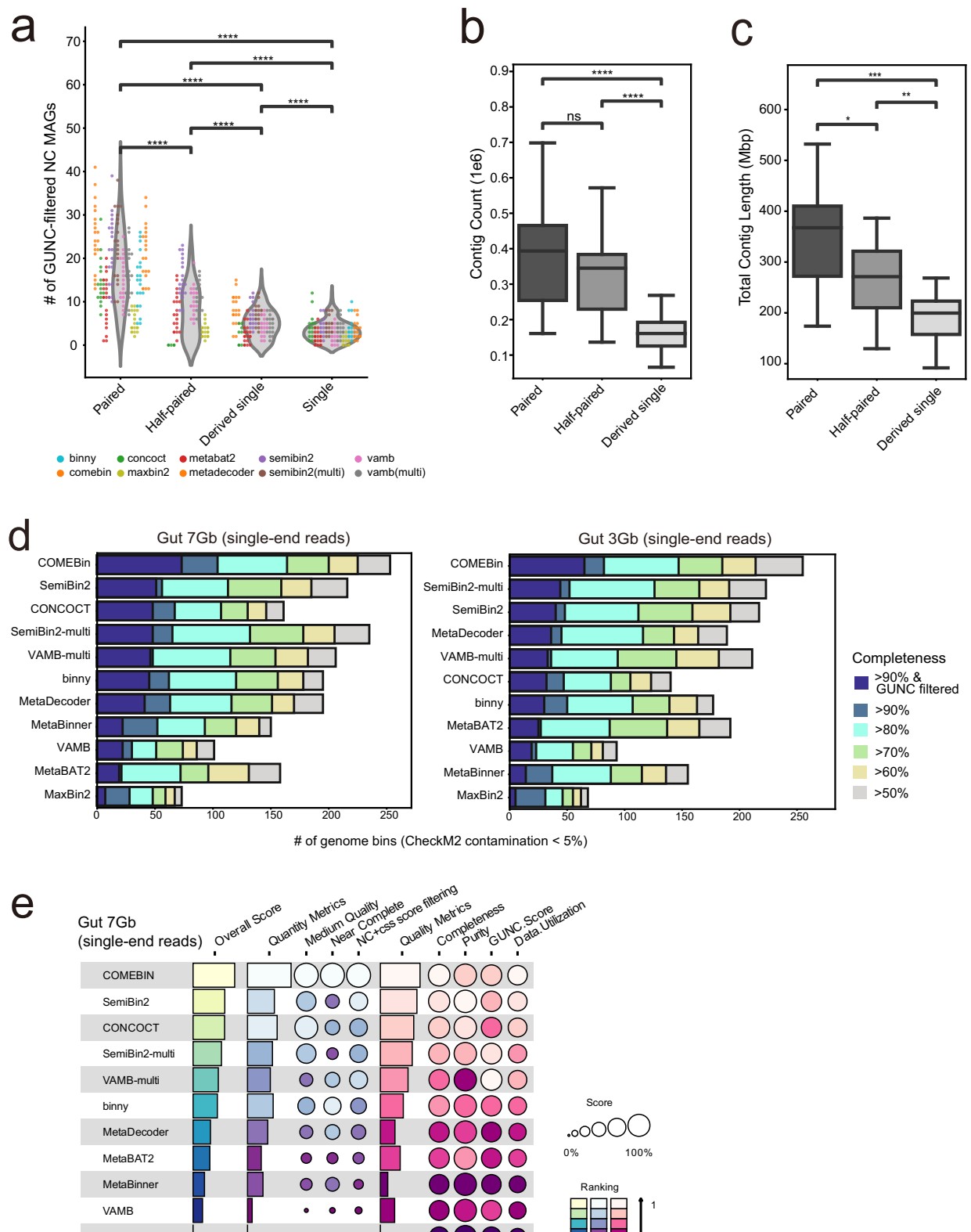

MetaBAT2, which was used for HRGM[33], oceanDNA MAG[34], and SMAG[35] catalogs. Using both custom simulated datasets and real human gut, ocean, and soil metagenomic datasets, our proposed ensemble binning pipeline consistently recovered substantially more genome bins than these established ensemble pipelines (Fig. 7c and Supplementary Fig. 4a–d). For example, on the simulated dataset c1000_10Gb, the proposed pipeline recovered 891 NC genome bins, whereas the next

best ensemble pipeline, MetaWRAP (CONCOCT, MaxBin2, MetaBAT2), recovered 673 NC genome bins—representing a 32.4% improvement. Similarly, for the real dataset Gut 7.2 Gb, the proposed pipeline recovered 599 NC genome bins, a 57% increase compared to the same runner-up ensemble pipeline, which recovered 381 NC genome bins. Notably, using a single neural network-based binner alone, such as COMEBin, VAMB-multi, or SemiBin2-multi, often yielded significantly

**Fig. 6 | Investigation of impact of the sequencing read formats on binning performance and benchmarks binning tools for single-end sequencing metagenomic data. a** GUNC-filtered near-complete (NC) MAG counts across paired-end, half-paired, derived single-end and single-end datasets (sequencing depth ≥5 Gb), shown as a combined violin plot (points coloured by tool). Individual points representing each tool are colour-coded to distinguish tool-specific trends. **b, c** Assembly quality metrics across read formats (contig count in (**b**); total contig length in (**c**); $n = 20$ samples per group). **d** Benchmarking results for metagenomic binning tools for single-end sequencing data in the human Gut 7 and 3 Gb datasets. Performance is measured by the number of genome bins with contamination under 5%, categorized by completeness thresholds of 50, 60, 70, 80, and 90%, with an additional GUNC filtration (CSS score <0.45). Completeness categories and GUNC-filtered bins are indicated by colour (see key in (**d**)). **e** Summary plot of the benchmarking results for the Gut 7 Gb single-end sequencing data (circle size indicates score; colour indicates ranking; see key in (**e**)). Statistical significance for (**a–c**) was assessed using the two-sided Mann–Whitney $U$ test (stars indicate unadjusted $P$ values; exact $P$ values are provided in the Source Data, with Benjamini–Hochberg-adjusted $P$ values reported for reference; *$p \leq 0.05$, **$p \leq 0.01$, ***$p \leq 0.001$, ****$p \leq 1.00 \times 10^{-4}$). Box lengths represent the interquartile range (IQR) of the data, and whiskers extend to the lowest and highest values within $1.5 \times$ IQR from the first and third quartiles, respectively. The center line indicates the median, and all outliers are shown.

more genome bins than the runner-up ensemble pipeline, MetaWRAP (CONCOCT, MaxBin2, MetaBAT2) in both custom simulated datasets and real human gut, ocean, and soil metagenomic datasets (Supplementary Fig. 4a–d).

Some genomic regions, such as 16S ribosomal RNA sequences, present particular challenges for genome recovery from metagenomes. We found that the proposed pipeline generally recovered more genomes containing 16S rRNA sequences than other ensemble pipelines (Fig. 7d). Furthermore, the proposed pipeline exhibited no detectable taxonomic bias in genome recovery, as the additional genomes obtained were evenly distributed across the phylogenetic tree (Fig. 7e). This indicates that the benefits of the updated pipeline extend beyond simply increasing the number of recovered genomes, as it also contributes to expanding the phylogenetic breadth of recovered genomes. Quantitative analysis of the taxonomic tree further confirmed a notable increase in the number of taxa across taxonomic ranks, with the most substantial increase observed at the species level (Fig. 7f). Together, these findings demonstrate that the newly proposed ensemble binning pipeline not only improves genome recovery but also enhances the capture of taxonomic diversity.

## Discussion

This study provides key insights into the performance of metagenomic binning tools and refinement pipelines for recovering high-quality MAGs, based on benchmarking analyses using both custom-simulated and real metagenomic datasets. We found that the commonly used CAMI-simulated datasets exhibit substantially lower taxonomic complexity compared to real host-associated and environmental microbial communities. Moreover, these CAMI datasets were generated with similar sequencing depths, making it difficult to assess the impact of sequencing depth on binning performance. To address these limitations, we generated custom simulated datasets with varying sequencing depths and taxonomic complexities.

Our analyses using these custom datasets demonstrated that binning performance correlates positively with sequencing depth and inversely with taxonomic complexity. While taxonomic complexity is an inherent characteristic of a microbial community, our results suggest that its negative impact can be partially mitigated by increasing sequencing depth, particularly when using neural network-based binning tools such as COMEBin, SemiBin2, SemiBin2-multi, and VAMB-multi. This highlights that dedicating additional effort and resources, such as performing ultra-deep sequencing combined with neural network-based binning tools, can substantially enhance genome recovery from complex metagenomes.

The overall results of this study suggest that neural network-based binning tools are the preferred choice for microbial genome reconstruction from metagenomic samples. Tools such as SemiBin2, COMEBin, and VAMB represent a paradigm shift from conventional clustering approaches by learning latent representations that capture complex, nonlinear relationships among contig features. In these frameworks, models are trained de novo for each dataset directly from contig composition and coverage patterns, enabling adaptive modeling without reliance on externally pretrained weights. This dataset-specific training allows the models to flexibly accommodate diverse microbial communities without explicit transfer learning.

However, because the models are trained within each dataset, their performance can be affected by data characteristics such as coverage heterogeneity, sequencing depth, and the presence of novel taxa. Moreover, neural network-based binning typically requires greater computational resources and yields latent embeddings that are less interpretable than the feature-based representations used in traditional clustering. The biological meaning of these learned representations is often not straightforward to assess, rendering such models relatively opaque compared to approaches relying directly on k-mer frequency and coverage. Thus, while neural network-based methods offer improved flexibility and accuracy, their efficiency, interpretability, and stability remain dependent on the quality and complexity of the input data.

SemiBin2 and VAMB, in particular, offer multi-sample binning options that enhance binning performance when integrating coverage across approximately 20 samples. Notably, VAMB demonstrated substantial performance gains with multi-sample binning. The benefits of using multi-sample coverage declined when more than 20 samples or fewer than 10 samples were used. The underlying reason for this diminishing return remains unclear. Consequently, we recommend employing multi-sample binning when available, but advise against using too few or too many samples to optimize performance. Another critical factor influencing binning efficacy was the sequencing read mode, single-end versus paired-end sequencing reads. We observed consistently poor binning performance for real metagenomic samples sequenced with single-end sequencing reads, regardless of the binning tool used. This poor performance was linked to inferior assembly quality, resulting in shorter contigs compared to assemblies generated from paired-end sequencing reads. These findings highlight the importance of using paired-end sequencing for genome-resolved metagenomics and strongly support its recommendation in future studies.

From our benchmarking study, we conclude that microbial genome recovery from metagenomic sequencing samples can be substantially improved through the following strategies. First, we recommend paired-end sequencing over single-end sequencing to improve binning efficacy by enhancing contig quality. Second, to overcome the high taxonomic complexity typical of real metagenomic samples, we suggest sequencing at least 7 Gb or more per sample. Third, when available, multi-sample binning modes should be utilized, particularly for VAMB, which showed substantially improved performance in this mode. However, we recommend using about 20 samples for optimal performance when performing multi-sample binning and avoiding using too few or too many samples to prevent potential performance degradation. Neural network-based binning tools consistently recovered more high-quality MAGs across varying sequencing depths and taxonomic complexities, with COMEBin emerging as the top choice for soil metagenomes, which present exceptionally high taxonomic complexity. Finally, we strongly recommend employing an

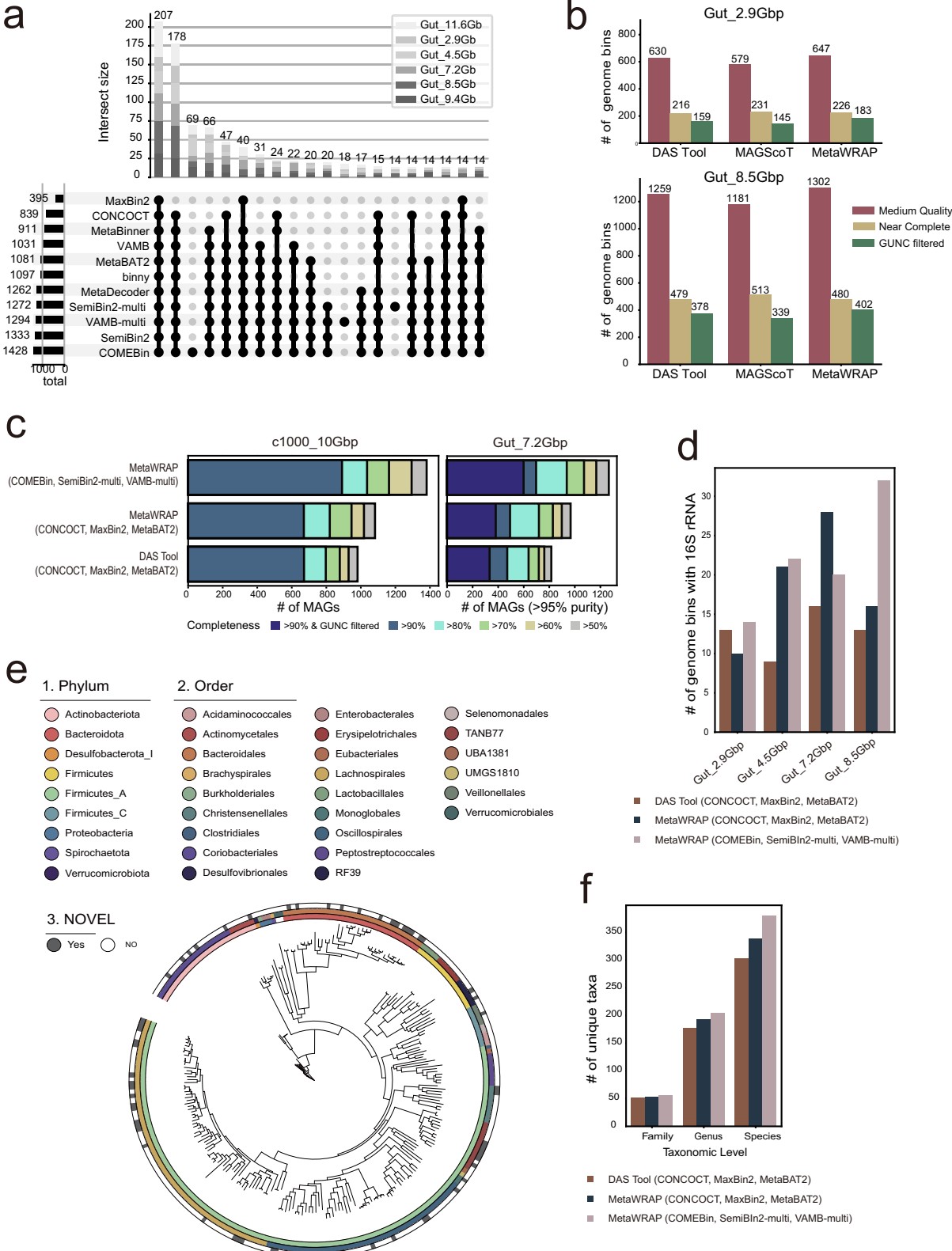

ensemble approach, combining multiple binning tools rather than relying on a single method, to maximize genome recovery and overall binning accuracy.

Our benchmark study primarily focused on evaluating binning tools in terms of genome recovery performance. However, improved recovery was often achieved through computationally intensive algorithms, such as deep neural network-based representation learning

(e.g., COMEBin, SemiBin2, VAMB). This trade-off between yield and efficiency should be carefully considered when selecting binners for real-world metagenomic projects. Although lightweight binners such as MetaBAT2 generally recover fewer genomes, they remain highly practical for large-scale analyses with limited computational resources.

We also note that marker gene–based iterative refinement contributes substantially to the computational overhead observed in some

**Fig. 7 | Performance and taxonomic diversity of the proposed ensemble binning pipeline. a** Genomic clustering of medium quality (MQ) MAGs or higher from all binning tools, performed at the species level using dRep with a 95% ANI threshold and a 80% minimum alignment coverage (see "Methods" for details). **b** Comparison of MQ MAGs, near complete (NC) MAGs, and NC MAGs filtered with GUNC (CSS score <0.45) among the three bin refinement tools, DAS Tool, MAGScoT, and MetaWRAP, in the Gut_2.9Gbp and Gut_8.5Gbp datasets (MQ, red; NC, yellow; GUNC-filtered NC, green). **c** Comparison of MAG counts generated by the proposed pipeline, pre-existing MetaWRAP, and DAS Tool pipelines in the c1000_10Gbp simulation and gut_7.2Gbp datasets. All MAGs with contamination under 5% were included and grouped by completeness thresholds of 50, 60, 70, 80,

and 90%. For real datasets, an additional category includes MAGs that meet the 90% completeness threshold and pass the GUNC filter (CSS score <0.45). **d** Detection rates of 16S rRNA regions recovered by three ensemble binning pipelines (colours denote pipelines). **e** A phylogenetic tree constructed using NC-level bins from the binning of Gut_7.2Gbp dataset using three ensemble binning pipelines. The resulting tree was visualized with iTOL, with the outermost strip highlighting novel taxa detected exclusively by the proposed pipeline, MetaWRAP (COMEBin, Semi-Bin2-multi, VAMB-multi). Inner color strips represent annotations at the phylum and order levels. **f** Quantitative analysis of taxonomic diversity at different taxonomic ranks (phylum, genus, species) based on the phylogenetic tree constructed in (**e**) (colours denote pipelines, as in (**d**)).

modern binners (e.g., binny, MetaBinner). Furthermore, marker-driven strategies, rather than marker-guided learning, may introduce lineage-specific biases. Therefore, future developments should aim to reduce such dependencies to ensure that performance improvements do not come at the cost of phylogenetic inclusivity.

Recently, a similar set of metagenomic binning tools was evaluated on real metagenomic datasets[20]. Our study distinguishes itself from the previous one in several ways. First, we assessed binners on both real and simulated datasets, which allowed us to investigate the factors influencing binning efficacy under more controlled conditions. Using these simulated datasets, we demonstrated that sequencing depth and taxonomic complexity strongly impact binning performance. Second, unlike the previous study, which did not consider genome chimerism, we found that chimeric genome rates vary widely across binning tools. For instance, although MetaBinner was selected as one of the top three high-performance binners in short read multi-sample binning in the previous study, it showed one of the highest rates of chimeric genomes. This suggests that MetaBinner may not be an ideal tool due to its high non-redundant genome contamination. Third, although the advantages of multi-sample binning have been demonstrated previously[29], we find that using either too few or too many samples can reduce these benefits. Fourth, we proposed an optimal ensemble binning pipeline and quantitatively demonstrated its advantages in genome recovery and taxonomic breadth, thereby reinforcing the practical utility of our approach for future metagenomic studies.

This study has several limitations. First, the simulation datasets were designed to achieve high taxonomic complexity by incorporating NCBI bacterial genomes from diverse environments. While this approach ensures broad diversity, it does not specifically mimic any single environmental condition, which may limit its direct applicability to certain real-world ecosystems. However, the inclusion of real environmental datasets and the CAMI human dataset, which yielded consistent results, helps mitigate this limitation. Second, the real environmental datasets were compiled from multiple independent projects with varying sequencing depths, each subject to different technical protocols and biological variations. These inconsistencies may obscure potential correlations between sequencing depth and binning performance in real metagenomic datasets. Third, some tools (e.g., COMEBin and MetaBinner) use single-copy marker genes (SCGs) to select a final binning output among multiple candidates, which may partially overlap with marker-gene-informed evaluation on real datasets (e.g., CheckM2) and complicate strict benchmarking independence. Although we acknowledge this potential concern, we mitigate it by additionally evaluating performance on simulated datasets using ground-truth-based metrics (AMBER), which are independent of marker-gene-based quality estimates and therefore provide an orthogonal assessment unaffected by SCG-driven model selection.

## Methods
### Synthetic metagenomic datasets
Synthetic metagenomic datasets were obtained from the human datasets of the second CAMI round[18], representing human gut, oral,

and airway microbiomes. To further investigate the impact of sequencing depth and taxonomic complexity on metagenomic binning performance, we generated additional simulated datasets using CAMISIM[24] (v1.3.0) with a de novo community design. A total of 20 custom datasets were generated using bacterial isolate genomes from the NCBI reference genomes, incorporating four levels of sequencing depth and five levels of taxonomic complexity. Each dataset consisted of 10 samples, with each sample representing a distinct microbial community generated with different combinations of sequencing depth and complexity levels. For each sample, 150-bp paired-end reads were generated using CAMISIM's default parameters with the Illumina MBARC error profile. Gold standard assembly contig files, which mitigate errors introduced during typical assembly processes, and read mapping files against all reference genomes were provided by CAMI or generated by CAMISIM. For multi-sample binning workflows, sample-specific contigs were concatenated using the "concatenate_fasta" script provided by SemiBin. A total of 30 sample reads from the CAMI datasets and 200 sample reads from the custom simulation datasets were aligned to their corresponding gold standard assembly contigs or concatenated contigs using Bowtie2 (v2.5.3).

### Real metagenomic datasets
We collected the majority of real metagenomic data from human gut microbiome samples. A total of 160 samples from 7 datasets were curated, with average sequencing depths ranging from 2.9 to 11.6 Gb (Supplementary Tables 2 and 3). Among these, six datasets consisted of paired-end reads, while one dataset consisted of single-end reads. Quality control was performed using Trimmomatic[36] (v0.39) to remove adapter sequences and low-quality (LQ) reads. Host-derived reads were filtered by aligning sequences to the human reference genome (GRCh38.p13) using Bowtie2[37] (v2.5.3). Each single-end sample was assembled using MEGAHIT[38] (v1.2.9), while paired-end samples were assembled using metaSPAdes[39] (v3.13.0). Read mapping files against either sample-specific contigs or concatenated contigs were generated using Bowtie2. Additionally, ocean and soil datasets were selected from the oceanDNA MAG[34] and SMAG[35] catalogs, respectively, and underwent the same processing steps, except for the host DNA removal step.

### Metagenomic binning and bin refinement
Initial binning results were produced using 9 genome binners, including binny[7] (v2.2.15), COMEBin[15] (v1.0.4), CONCOCT[6] (v1.1.0), MaxBin2[8] (v2.2.7), MetaBAT2[10] (v2.15), MetaBinner[16] (v1.4.4), MetaDecoder[11] (v1.0.16), SemiBin2[14] (v1.4.0), and VAMB[12] (v3.0.9). Most programs were run with their default settings, but VAMB was configured with the recommended setting (-m 2000) and SemiBin2 was conducted with the "–self-supervised" option. Bin refinement step was produced using best binning results, retrieved through MetaWRAP refinement module[21] (v1.3.2), DAS Tool[22] (v1.1.6), and MAGScoT[23] (v.1.0.0). We modified the MetaWRAP code to select the genome bin with the highest quality based on CheckM2[26] (v0.1.2) evaluations, replacing the previous assessment methods that relied on CheckM[40].

## Assessment of genome bin quality

For the CAMI and custom simulation datasets, we used the AMBER[25] (Assessment of Metagenome BinnERs) package (v2.0.0) to evaluate genome binning results. To assess each genome bin, AMBER identifies the gold standard genome with the highest proportion of shared base pairs within that bin and assigns it as the reference genome. Metrics, including completeness, purity, and ARI, were calculated. Additionally, average purity per base pair and completeness per base pair were computed to assess performance in recovering genomes across different abundance levels. These metrics were used to compare the effectiveness of various metagenomic binning tools. For real metagenomic datasets, CheckM2 was used to evaluate completeness and contamination. Chimeric genomes were then detected using GUNC[28] (v1.0.6) with default thresholds. All prokaryotic genome bins were subsequently classified into three predefined quality tiers, with an additional criterion for chimerism (GUNC-passed): LQ (completeness <50% and contamination ≥5%), medium-quality (MQ: completeness ≥50% and contamination <5%), and NC (completeness ≥90%, contamination <5%, and GUNC-passed).

## Summarizing benchmarking results

To rank the binning tools, we constructed a benchmark summary figure that aggregates key performance metrics across various datasets using a heatmap visualization[41]. The metrics were divided into two main categories: quantity metrics (e.g., the number of genome bins meeting specific quality thresholds) and quality metrics (e.g., average purity and completeness). Quantity metrics were standardized across datasets using min-max scaling to ensure equal weighting. In contrast, quality metrics, which typically range between 0 and 1, were not scaled to avoid exaggerating minor variations, for instance, differences of only 0.03 in completeness or 0.1 in precision. Both categories were given equal weight when calculating the final scores, and the tools were ranked based on their overall performance across all datasets.

## Evaluation of computational efficiency of binning and refinement tools

Computational efficiency across binning and refinement tools was evaluated on a Linux server equipped with two AMD EPYC 7702 processors (64 cores each, 2.0–3.3 GHz) and 512 GB of system memory, using five CPU cores per sample of the Gut 7.2 Gb dataset. GPU-accelerated tools (e.g., COMEBin-GPU) were executed on a dedicated computational node equipped with six NVIDIA RTX A6000 GPUs (48 GB VRAM each; CUDA v13.0), using a single GPU per sample of the Gut 7.2 Gb dataset.

## Generation of species clusters and taxonomic annotations

Genome clustering was performed at the species level for genome bins classified as MQ and NC by all binning tools using a two-step clustering process implemented in dRep[31] (v3.4.5). In the initial step, preliminary clustering was conducted using Mash[42] (v2.2) to rapidly calculate all-by-all genomic distances. In the second step, ANI was computed for genome pairs within each preliminary cluster using fastANI[43] (v1.33) to refine the clustering accuracy. Species-level clusters were defined using a 95% ANI threshold with a minimum alignment coverage of 80%, ensuring robust genome comparisons. Within each cluster, we selected a genome with a highest intactness score $(S)$[32,33], $S = Completeness − 5 × Contamination + 0.5 × \log_{10}(N50)$, as the representative genome. To examine the uniqueness and overlap of species clusters across different binning tools, we used Upset[44] visualization. For phylogenetic analysis, clustering was restricted to NC genome bins. Representative genomes were taxonomically annotated using GTDB-Tk[45] (v2.1.1), and the resulting multi-sequence alignment file was used to construct a phylogenetic tree with IQ-TREE[46] (v2.3.6). The tree was visualized in iTOL[47], providing a comprehensive view of the taxonomic and evolutionary relationships.

## Reporting summary

Further information on research design is available in the Nature Portfolio Reporting Summary linked to this article.

## Data availability

All benchmarking data is publicly available. The simulated benchmarking data generated in this study have been deposited in the Zenodo database under accession code https://doi.org/10.5281/zenodo.18916168[48]. The processed benchmarking results data are available at https://github.com/netbiolab/Benchmarking_metagenomic_binners[49]. The CAMI II Toy Human Microbiome Project data used in this study are available in PUBLISSO with DOIs 10.4126/FRL01-006425518 and on the CAMI data portal (https://data.cami-challenge.org/participate). The real metagenome datasets used in this study are available in the European Nucleotide Archive (ENA) under accession code PRJNA530339, PRJNA797994, PRJEB27005, PRJEB33013, PRJEB39223, PRJDB4525, PRJNA624763, PRJNA273799, PRJNA261849. All data supporting the findings described in this manuscript are available in the article and in the Supplementary Information. Source data are provided with this paper.

## Code availability

All commands and code used for processing and analyzing the data are publicly available at https://github.com/netbiolab/Benchmarking_metagenomic_binners[49].

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

## Acknowledgements

This work was supported by the National Research Foundation (NRF) funded by the Ministry of Science and ICT, Republic of Korea (2022M3A9F3016364, 2022R1A2C1092062, and RS-2025-18362970 to I.L.), and Brain Korea 21 (BK21) FOUR program.

## Author contributions

J.K. and I.L. conceived and designed the study. J.K. conducted benchmarking analysis of metagenomic binning methods under supervision of I.L., N.K., J.H.C., and J.M. provided technical advice and support. J.K. and I.L. wrote and edited the manuscript.

## Competing interests

I.L. is a founder of and shareholder in DECODE BIOME. The other authors declare no competing interests.
