## [Transparent Peer Review file · Nature Communications]

Comprehensive benchmarking of metagenomic binning tools reveals key factors for improved genome recovery

Corresponding Author: Professor Insuk Lee

Version 0:

Reviewer comments:

Reviewer #1

(Remarks to the Author)

Jungyeon Kim et al presented comprehensive evaluations of metagenomic binning tools for better genome recovery from shotgun metagenomics dataset. Not only using simulated dataset, but also real metagenomics dataset supported the validity benchmarking result for assessing binning tools. Checking genomic chimerism was performed properly by GUNC, which is timely review of the metagenomic binning tools as currently many MAGs were actually mixed genomes from the closely related strains, thereby boosting the number of MAGs detected. This reviewer has few minor comments to improve the clarity of the manuscript.

1) Currently many tools were checked, but somehow its categorization is not clear. for example, it was mentioned as "traditional" or "neural network based" for some tools, and "traditional" looks unclear to me. is it possible to have clear categorizations of the tools based on their algorithmic basis or other rationale?

2) evaluating computational resources used during metagenomic binning is important and shown in supple table 5. how about same evaluations for the ensembl refinement tools, such as DAS tools, metaWRAP? it is good to check how much time will be spent for each tool

3) it seems not all evaluations of refinement tools were checked for same metagenomic tools. For example, how's the performance of DAS tool using COMEBin, SemiBin2-multi, and VAMB-multi binning result? this combination was checked with metaWRAP, but it seems not checked by DAS tools (Figure 6 c,d)

4) checking binning quality by single-end sequencing vs paired end sequencing looks interesting. how about long read sequencing? it would be nice to check binning quality comparing single end vs long read vs paired end

5) mock community can be good dataset to consider for evaluations of metagenomic binning quality if possible. (<https://www.nature.com/articles/s41597-019-0287-z>)

6) authors claimed that figure 3a showed correlations of binning quality and taxonomic complexity and sequencing depth. how significant is this correlation?

7) in general, this paper claimed that neural network-based tools outperformed than other tools. I guess this conclusion can be changed if evaluations were done by dataset not used for training the neural networks of given tools. discussions of pros and cons of neural network-based tools can be interesting

8) is it possible to detect MAGs of non-bacteria organisms, such as fungi or virus, and also possible to identify mobile genetic elements (plasmids, transposon, CRISPR-Cas, etc) from the high-quality MAGs ? this can be interesting subject in this metagenomic binning quality assessments

Reviewer #2

(Remarks to the Author)

The authors conduct a comprehensive benchmark of metagenomic binning tools and later refinement tools using a combination of simulated and real metagenomic datasets. The study evaluates the effect on binning performance given variations in sequencing depth, taxonomic complexity, number of samples used in multi-sample binning, and single-end vs. paired-end sequencing. The authors find that (similarly to already reported findings) more modern, neural network-based, binners consistently outperform other binners in quality, especially in datasets with high taxonomic complexity. The study also highlights the issue of genomic chimerism, which appears to vary between tools and is not captured by standard contamination (CheckM) assessments. Later, as expected from previous literature, the authors also show that multi-sample binning is most effective. New data from the authors specify that with approximately 20 samples the binning yield does not improve for multi-sample binning projects. Finally, based on their findings, the authors propose an optimized ensemble pipeline combining three top-performing neural network-based tools (COMEBin, SemiBin2-multi, VAMB-multi) with the MetaWRAP refinement module, showing it recovers over 30% more high-quality genomes than established methods.

1. While the manuscript presents a valuable contribution of higher quality than other recent benchmarking efforts published in this journal a few weeks ago, a primary concern is its practical applicability to the future of metagenomics, which is defined by an increasing number of samples and data volume. The top-performing binners identified by the authors are extremely inefficient; for instance, Supplementary Table 5 shows that for the Gut 7.2Gb dataset, binny required over three days and COMEBin over 13 days to complete, a critical limitation the authors only vaguely mention in the manuscript. A scientifically fair benchmark must be clear on how realistic a tool is for the current needs of the field. Claiming inefficient binners are the best is not appropriate without clearly listing their limitations, especially when a go-to binner for large-scale projects, MetaBAT2, was not included in the resource analysis in Supplementary Table 5. There is a very practical reason as to why MetaBAT2 is still the preferred binner in large-scale projects, for example see <https://www.nature.com/articles/s41587-020-0718-6>, <https://www.nature.com/articles/s41586-024-08240-z>, [https://www.cell.com/cell/fulltext/S0092-8674\(24\)00833-X](https://www.cell.com/cell/fulltext/S0092-8674(24)00833-X), <https://www.biorxiv.org/content/10.1101/2024.06.27.600767v1>. The authors should be more explicit about what "consistently outperformed others" means in this context.

2. Both, this and the recently published benchmark, seem to narrowly focus on binners that yield more high-quality bins, which, in the case of binners that iteratively look for marker genes to refine MAGs, may bias results and neglect lineages discoverable by more unbiased methods. Some of the modern binners appear to be closer to refinement tools than to actual unbiased, marker-gene agnostic binning tools. The authors should be more explicit in recognizing a key scientific gap they found in their analysis: the trade-off between modern binners that take days to yield a percentage increase in MAGs versus highly efficient binners that complete in minutes; so the field of metagenomics can move in the right direction with the future of binning tools, and avoid investing/wasting disproportional amounts of computing resources and energy consumption when using inefficient tools that yield small gains in number of MAGs compared to the computing resource they demand.

3. The authors identify genomic chimerism as an often overlooked aspect of bin quality and employ the tool GUNC to assess this non-redundant contamination. Their analysis shows that the proportion of chimeric genomes varies between different binning tools, which is an interesting finding that highlights the limitations/variability of relying solely on marker-gene-based assessments. However, I think that caution should be exercised when interpreting GUNC's output. Based on our own thorough testing of this tool, we have found that it can greatly overestimate contamination and the detection of chimeras. Therefore, while the relative differences in chimerism rates across the various binners are an important takeaway from this study, I would be a little more conservative with the wording around the interpretation of GUNC's results.

4. The current framing of the results directly compares binning outcomes from single-end versus paired-end sequencing, concluding that paired-end sequencing yields superior results. However, this comparison is less a benchmark of the binning tools themselves and more a reflection of upstream data quality and processing steps, particularly the performance of the contig assembler. The observed reduction in binning efficacy for single-end samples is an expected consequence of poorer contig quality and increased assembly fragmentation, as the assemblers are limited by the lack of spatial linkage information. To maintain the manuscript's focus on the performance of the binning algorithms, I would suggest that the authors reframe this section. Instead of comparing the two sequencing formats directly, the analysis should compare the performance of the different binners within each format. This approach would answer more relevant questions, such as, "Which binning tools are most effective at recovering genomes from the lower-quality, fragmented assemblies typical of single-end data?" and "Which tools perform best given high-quality paired-end assemblies?" This reframing would provide a more direct and valuable assessment of the binners' robustness and capabilities under different input conditions.

5. The manuscript contains a factual error, stating CheckM2 relies on single-copy marker genes, which was true for CheckM1 but not the machine-learning-based CheckM2.

6. The usage of AI tools for writing is evident in the style and grammar used. I am not against it at all, but the tone of the paper starts to sound very homogeneous and not building excitement as the manuscript progresses. The authors should at least acknowledge and disclose their use of AI tools, and the specific models used in this case.

All in all, I consider this manuscript to be of high relevance, and I will be more than happy to review it again provided the authors revise it accordingly. The paper is a valuable contribution that I believe will help guide the design of future metagenomic studies.

Reviewer #3

(Remarks to the Author)

This paper benchmarks binning algorithms for reconstructing bacterial genomes from metagenomic datasets, using commonly used datasets, newly simulated datasets, and real metagenomic datasets. The authors report several notable findings, including that newer neural network–based methods outperform earlier approaches. They also propose a new pipeline that integrates and refines genome bins from the top three tools. This pipeline recovered over 30% more high-quality genomes than previous methods.

The paper offers practical guidance for improving metagenomic binning techniques. However, some findings are expected and not particularly novel—for example, that sequencing depth and taxonomic complexity strongly influence performance.

More detail is needed on the newly simulated datasets. Simply stating that "CAMISIM24 (v1.3.0) with a de novo community design" and "default parameters was used" is insufficient. Information about the microbial community complexity (e.g., gut, oral, or other environments), taxonomic composition, and abundance distributions represented in the simulated datasets would be valuable.

Certain aspects of the benchmarking could also be strengthened. For example, the paper concludes that multi-sample binning is most effective with around 20 samples, as too few or too many samples reduce its benefits. However, this conclusion appears to be based on a single collection of simulated datasets. It remains unclear whether this result would hold under different microbial community complexities or sequencing depths.

Version 1:

Reviewer comments:

Reviewer #2

(Remarks to the Author)

1. I do not agree that COMEBin (or MetaBinner) should be benchmarked together with real binning tools. COMEBin uses CheckM marker genes to build/cluster bins, COMEBin pre-filters binning results using CheckM, and then the benchmarking is done using CheckM... it is just not an independent or fair way to compare binning performance, because the method is effectively being evaluated with the same marker framework it is built around.

2. The authors addressed all my comments.

Reviewer #3

(Remarks to the Author)

The authors have fully addressed my concerns in this revision.

Version 2:

Reviewer comments:

Reviewer #2

(Remarks to the Author)

The authors have addressed all my comments. Thanks!

Responses to referees (our responses in bold or colored text)

Manuscript ID: NCOMMS-25-34019-T

Attached please find our revised manuscript. We do appreciate the reviewers' critical but valuable comments, which have been a guideline in revising the manuscript. We have provided a revised manuscript, Figures (with source data), and Supplementary tables. We believe that the revisions improved the manuscript particularly in delivering our analysis to the readers clearly. We hope that our revised manuscript is suitable for the publication.

=====

REVIEWER COMMENTS

Reviewer #1 (Remarks to the Author):

Jungyeon Kim et al presented comprehensive evaluations of metagenomic binning tools for better genome recovery from shotgun metagenomics dataset. Not only using simulated dataset, but also real metagenomics dataset supported the validity benchmarking result for assessing binning tools. Checking genomic chimerism was performed properly by GUNC, which is timely review of the metagenomic binning tools as currently may MAGs were actually mixed genomes from the closely related strains, thereby boosting the number of MAGs detected. This reviewer has few minor comments to improve the clarity of the manuscript.

1) Currently many tools were checked, but somehow its categorization is not clear. for example, it was mentioned as "traditional" or "neural network based" for some tools, and "traditional" looks unclear to me. is it possible to have clear categorizations of the tools based on their algorithmic basis or other rationale?

We apologize that we referred to widely used tools as “traditional tools,” which was ambiguous. We have replaced “traditional tools” with “previously widely used tools” for clarity in the revised manuscript.

In the initially submitted manuscript, we explicitly describe **four algorithmic categories based on how composition and abundance features are integrated prior to clustering**: projection-based, probabilistic, neural network-based, and ensemble-based models.

(Line 57) *“Nevertheless, integrating heterogeneous features and clustering them remains a significant challenge, leading to the development of numerous approaches. These algorithms can be broadly classified into four categories based on their machine learning model structures, especially how these features are processed and integrated prior to clustering (Fig. 1a).”*

To improve description of the categories, we have revised Supplementary Table 1.

2) evaluating computational resources used during metagenomic binning is important and shown in supple table 5. how about same evaluations for the ensembl refinement tools, such as DAS tools, metaWRAP? it is good to check how much time will be spent for each tool

We appreciate the reviewer's suggestion. We re-evaluated runtime and memory usage for DAS Tool, MetaWRAP, and MAGScoT in 5 CPU core configuration using the same combination of binners (COMEBin, SemiBin2-multi, and VAMB-multi). The results have been added to **Supplementary Table 6**. Among the three tools, MAGScoT required the shortest runtime (~2.5

min per sample), followed by DASTool (~8 min per sample), while MetaWRAP showed the highest computational cost (~85 min per sample). However, considering that MetaWRAP provides significantly more high-quality genomes, and that a well-established reference genome catalog can serve many studies for years, we believe that such computational investment is justified. Therefore, we proposed an optimal binning pipeline with refinement using MetaWRAP. We have added these descriptions and discussions in the revised manuscript as follows:

(Line 397) *“We further evaluated computational efficiency of bin refinement tools using a 5-core CPU configuration and found that MetaWRAP required considerably longer runtime (~85 min per sample) than DAS Tool (~8 min per sample) and MAGScoT (~2.5 min per sample) (Supplementary Table 6). Nevertheless, considering that a well-curated genome catalog can serve numerous downstream analyses for years, we believe that this computational investment can be justified when sufficient CPU resources are available (e.g., refinement of 1,000 samples can be completed in 3 days with 100 CPU cores). Focusing primarily on maximizing genome recovery, we propose an optimal ensemble binning pipeline that combines COMEBin, SemiBin2-multi, and VAMB-multi, followed by bin refinement using MetaWRAP.”*

3) it seems not all evaluations of refinement tools were checked for same metagenomic tools. For example, how's the performance of DAS tool using COMEBin, SemiBin2-multi, and VAMB-multi binning result? this combination was checked with metaWRAP, but it seems not checked by DAS tools (Figure 6 c,d)

As suggested, we have performed bin refinement using DASTool, MAGScoT, and MetaWRAP on the results from COMEBin, SemiBin2-multi, and VAMB-multi and summarized performance based on genome recovery (Fig. 7b) in the revised manuscript as follows:

(Line 390) *“To evaluate the effectiveness of bin refinement, we compared a commonly used MetaWRAP refinement module and DAS Tool with a more recently developed tool, MAGScoT, using real human gut metagenome datasets with varying sequencing depths (Gut 2.9 and 8.5 Gb). Across both datasets, MetaWRAP consistently outperformed DAS Tool and MAGScoT in the recovery of GUNC-filtered NC genome bins (Fig. 7b). Notably, while DAS Tool and MAGScoT retrieved comparable number of NC genomes before GUNC filtration, they produced significantly more chimeric genomes, ultimately resulting in fewer genome bins than MetaWRAP after GUNC filtration.”*

4) checking binning quality by single-end sequencing vs paired end sequencing looks interesting. how about long read sequencing? it would be nice to check binning quality comparing single end vs long read vs paired end.

Although we agree that benchmarking for long-read sequencing data is also important, **we intentionally focused this study on metagenomic binning using short-read-based assembled contigs, as most publicly available datasets have been generated by short-read sequencing.** Focusing on a single sequencing platform allowed us to conduct a systematic benchmarking analysis across various factors affecting efficacy of high-quality MAGs, including chimerism, sequencing depth, taxonomic complexity, read layout (single-end vs paired-end), and the number of samples used for multi-sample binning. Therefore, **including benchmarking results for long-read sequencing data would be beyond the scope of this study and could potentially dilute the clarity of our conclusions.**

In fact, for long-read-based metagenome assembly, sequencing accuracy and assembly quality are often more critical than binning performance, as long reads typically produce much longer but less accurate contigs at the nucleotide level. The recently developed HiFi sequencing technology can achieve accuracy comparable to short-read sequencing; however, due to its high cost, it will likely take time before becoming a standard platform in metagenomic research.

We deeply appreciate the reviewer’s insightful comment, as **we acknowledge that our rationale for focusing on short-read data was not clearly articulated in the original manuscript.** Accordingly, we have revised the text in the updated manuscript to clarify this point as follows.

(Line 18) *“We benchmarked various binning tools using CAMI-simulated, custom-simulated, and real metagenomic datasets, primarily focusing on short-read sequencing data.”*

(Line 112) *“Our objective is to establish a comprehensive and reliable benchmarking workflow for metagenomic binning tools, with a focus on short-read sequencing data that underpin most MAG cataloging projects.”*

5) mock community can be good dataset to consider for evaluations of metagenomic binning quality if possible.

(<https://www.nature.com/articles/s41597-019-0287-z>)

We agree that mock communities can serve as valuable benchmarking materials. A mock community is an intentionally constructed mixture of several microbial isolate strains at

defined proportions, offering well-characterized references for evaluation. As reviewer suggested, the ZymoBIOMICS Microbial Community Standard is a representative example, in which the reference genomes of all constituent strains are precisely known. Because it is a wet-lab-generated sample rather than a computational simulation, it offers a high degree of biological realism.

However, mock communities generally exhibit low taxonomic complexity, typically containing fewer than 100 strains across limited phyla. For example, the ZymoBIOMICS BMOCK12 mock community comprises only 12 strains for 4 phyla, while even the most complex synthetic mock set includes only 87 strains across 29 phyla. This raises concerns about their ability to represent the diversity encountered in real metagenomic samples. Indeed, most existing CAMI benchmarking efforts already cover such low- to medium-complexity communities.

To extend the benchmarking scope, we generated high-complexity synthetic datasets using CAMISIM (v1.3.0) that include hundreds to thousands of species with approximately log-normal abundance distributions. These datasets provide a standardized yet environment-agnostic framework **for assessing tool performance under realistic taxonomic diversity and sequencing conditions, which is our main goal in this study.**

6) authors claimed that figure 3a showed correlations of binning quality and taxonomic complexity and sequencing depth. how significant is this correlation?

We thank the reviewer for this question. **We have quantified these relationships using the Spearman correlation coefficient (ρ) and associated p-values.** The corresponding statistical plots have been added to **Supplementary Figure 1c** for clarity, and the Results section (line 212) has been updated as follows.

(Line 212) *“Most notably, completeness scores positively correlated with sequencing depth (Spearman correlation $\rho = 0.20$, $p < 7.6e-44$), while showing an inverse correlation with taxonomic complexity (Spearman correlation $\rho = -0.55$, $p < 1e-100$) across all binning tools (Fig. 3a, Supplementary Fig. 1c).”*

7) in general, this paper claimed that neural network-based tools outperformed than other tools. I guess this conclusion can be changed if evaluations were done by dataset not used for training the neural networks of given tools. discussions of pros and cons of neural network-based tools can be interesting

We appreciate the reviewer’s thoughtful comment and agree that, in general, neural network models can show reduced performance when evaluated on datasets that differ substantially from those used for training. This is particularly relevant for *pretrained* models, where learned representations may not generalize well to communities or environments with distinct compositional structures.

However, **the training process in neural network–based binning tools fundamentally differs from conventional machine learning paradigms such as pretraining or model transfer.** In typical supervised learning, models are trained on large external datasets and then applied to new data, where cross-dataset generalization becomes a key concern. In contrast, neural network–based bidders train models *de novo* for each new input dataset. **Here, “training” refers to the process of learning latent representations (embeddings) directly from the contig composition and coverage profiles of the dataset under analysis.** The resulting embedding space is immediately used for clustering (binning) on the same dataset, making the process entirely self-contained. **Consequently, cross-dataset bias or overfitting to external data does not arise,** unless a pretrained external model is explicitly applied.

The tools evaluated in our study, SemiBin2, COMEBin, and VAMB, all follow this dataset-specific training paradigm. **SemiBin2** performs self-supervised learning by automatically generating must-link and cannot-link constraints from the input data, thereby learning embeddings without external supervision. **COMEBin** employs contrastive multi-view representation learning, creating multiple augmented views of each contig within the same dataset to train the embedding network. **VAMB** uses a variational autoencoder (VAE) that jointly models sequence composition and abundance profiles derived from the same dataset. **In all cases, the models are trained anew for each dataset,** with no reuse of parameters or weights across datasets.

Therefore, **while the reviewer’s concern would be relevant for pretrained or transferred models, it does not apply to the dataset-specific, self-contained training schemes used here.** Instead, the primary limitation of neural network–based bidders lies in the stability and reproducibility of the learned embeddings, particularly when dealing with highly heterogeneous, low-coverage, or taxonomically novel datasets. As suggested, we have added more discussions about pros and cons of neural network-based bidders in Discussion section as follows.

(Line 457) *“Tools such as SemiBin2, COMEBin, and VAMB represent a paradigm shift from conventional clustering approaches by learning latent representations that capture complex, nonlinear relationships among contig features. In these frameworks, models are trained de novo for each dataset directly from contig composition and coverage patterns, enabling adaptive modeling without reliance on externally pretrained weights. This dataset-specific training allows the models to flexibly accommodate diverse microbial communities without explicit transfer learning.*

However, because the models are trained within each dataset, their performance can be affected by data characteristics such as coverage heterogeneity, sequencing depth, and the presence of novel taxa. Moreover, neural network–based binning typically requires greater computational resources and yields latent embeddings that are less interpretable than the feature-based representations used in traditional clustering. The biological meaning of these learned representations is often not straightforward to assess, rendering such models relatively opaque compared to approaches relying directly on k-mer frequency and coverage. Thus, while neural network–based methods offer improved flexibility and accuracy, their efficiency, interpretability, and stability remain dependent on the quality and complexity of the input data.”

8) is it possible to detect MAGs of non-bacteria organisms, such as fungi or virus, and also possible to identify mobile genetic elements (plasmids, transposon, CRISPR-Cas, etc) from the high-quality MAGs? this can be interesting subject in this metagenomic binning quality assessments

MAG assembly for fungal genomes is particularly challenging due to their larger sizes, repetitive content, intron-rich architecture, multiple chromosomes, and variable ploidy/heterozygosity. As a result, assembling fungal genomes from bulk metagenomic data is not routine, and the few published fungal MAGs are generally low quality.

By contrast, viral MAG assembly is an emerging field. Because the binning tools evaluated in our study were developed for prokaryotic genomes, their performance on viral contigs is suboptimal. This has prompted the development of virus-specific methods—e.g., vRhyme, PHAMB, CoCoNet, and Phables—and a recent benchmarking study (PMID: 39707560) presents comparative results for viral genome assembly/binning.

We agree that evaluating binning capability for fungal and viral genomes is an interesting direction. However, we did not include these analyses for two reasons: (i) the tools benchmarked here were not designed for fungal or viral genomes and are therefore unlikely to perform well; and (ii) our study focuses on prokaryotic genome assembly/binning, and adding fungal/viral evaluations would dilute the scope and conclusions.

To clarify that this study focused on metagenomic binning for reconstructing prokaryotic genomes, we have revised text in the revised manuscript as follows:

(Line 17) *“Metagenomic binning is essential for reconstructing **prokaryotic** genomes from metagenomic samples.”*

(Line 30) *“This study provides practical guidance for improving metagenomic binning **to facilitate the reconstruction of prokaryotic genomes.**”*

The MAG approach has limitations in recovering 16S rRNA regions and mobile genetic elements, not due to binning but to the inherent constraints of assembly. Consequently, even high-quality MAGs often lack complete coverage of these loci. To overcome these challenges, long-read sequencing (e.g., ONT or PacBio HiFi) is generally required.

Reviewer #2 (Remarks to the Author):

The authors conduct a comprehensive benchmark of metagenomic binning tools and later refinement tools using a combination of simulated and real metagenomic datasets. The study evaluates the effect on binning performance given variations in sequencing depth, taxonomic complexity, number of samples used in multi-sample binning, and single-end vs. paired-end sequencing. The authors find that (similarly to already reported findings) more modern, neural network-based, binners consistently outperform other binners in quality, especially in datasets with high taxonomic complexity. The study also highlights the issue of genomic chimerism, which appears to vary between tools and is not captured by standard contamination (CheckM) assessments. Later, as expected from previous literature, the authors also show that multi-sample binning is most effective. New data from the authors specify that with approximately 20 samples the binning yield does not improve for multi-sample binning projects. Finally, based on their findings, the authors propose an optimized ensemble pipeline combining three top-performing neural network-based tools (COMEBin, SemiBin2-multi, VAMB-multi) with the MetaWRAP refinement module, showing it recovers over 30% more high-quality genomes than established methods.

1. While the manuscript presents a valuable contribution of higher quality than other recent benchmarking efforts published in this journal a few weeks ago, a primary concern is its practical applicability to the future of metagenomics, which is defined by an increasing number of samples and data volume. The top-performing binners identified by the authors are extremely inefficient; for instance, Supplementary Table 5 shows that for the Gut 7.2Gb dataset, binny required over three days and COMEBin over 13 days to complete, a critical limitation the authors only vaguely mention in the manuscript. A scientifically fair benchmark must be clear on how realistic a tool is for the current needs of the field. Claiming inefficient binners are the best is not appropriate without clearly listing their limitations, especially when a go-to binner for large-scale projects, MetaBAT2, was not included in the resource analysis in Supplementary Table 5. There is a very practical reason as to why MetaBAT2 is still the preferred binner in large-scale projects, for example see <https://www.nature.com/articles/s41587-020-0718-6>, <https://www.nature.com/articles/s41586-024-08240-z>, [https://www.cell.com/cell/fulltext/S0092-8674\(24\)00833-X](https://www.cell.com/cell/fulltext/S0092-8674(24)00833-X), <https://www.biorxiv.org/content/10.1101/2024.06.27.600767v1>. The authors should be more explicit about what "consistently outperformed others" means in this context.

First of all, **we apologize for providing inaccurate information about the runtime of binning tools in our initial submission. We found that the runtime measurements were overestimated due to uncontrolled background processes running on the same server during the previous benchmarking.** We sincerely regret this oversight, which resulted from our primary focus on genome recovery performance rather than computational efficiency.

We fully agree that computational efficiency should be considered when providing guidance on the selection of binning tools for specific projects. Therefore, **we carefully re-evaluated the runtime** and memory usage of all tested tools, using five AMD CPU cores per sample of the Gut 7.2 Gb dataset or a single NVIDIA RTX A6000 GPU for GPU-accelerated tools (e.g., COMEBin-GPU), **ensuring that no background processes were active on the server.** This new runtime benchmarking process has been incorporated into the revised manuscript, as described below.

(Line 614) ***Evaluation of computational efficiency of binning tools***

Computational efficiency across binning tools was evaluated on a Linux server equipped with two AMD EPYC 7702 processors (64 cores each, 2.0–3.3 GHz) and 512 GB of system memory, using five CPU cores per sample of the Gut 7.2 Gb dataset. GPU-accelerated tools (e.g., COMEBin-GPU) were executed on a dedicated computational node equipped with six NVIDIA RTX A6000 GPUs (48 GB VRAM each; CUDA v13.0), using a single GPU per sample of the Gut 7.2 Gb dataset.”

This new runtime analysis yielded more accurate and realistic measurements, which have been summarized in **Figure 4e** of the revised manuscript as follows.

All resource measurements in our benchmark were conducted under a uniform and deliberately limited configuration of five CPU cores, providing a fair basis for comparison but representing a resource-constrained environment. Consequently, the reported runtimes should be interpreted as conservative (upper-bound) runtime estimates under limited resources, rather than optimal runtimes achievable with full hardware utilization.

We found that COMEBin and binny were particularly time-consuming. While MetaBAT2 required only 5 min to complete binning for each metagenomic sample, COMEBin, which consistently achieved the highest genome recovery, required 247.2 min (≈ 4.1 h). Therefore, as the reviewer correctly pointed out, such runtime may not be practical for MAG reconstruction projects involving hundreds or thousands of samples.

Fortunately, GPU-based servers are becoming increasingly common, and a GPU-accelerated implementation of COMEBin is now available. When run on a single NVIDIA RTX A6000 GPU, which is not a high-end model, the binning process can be completed within about one hour. The original COMEBin paper likewise benchmarked the GPU-enabled mode on a workstation equipped with two Intel Xeon E5-2678 CPUs (2.5 GHz) and one RTX 4090 GPU (48 threads). In that setup, the GPU version completed the dataset with 53 samples of > 250,000 contigs in about 6 hours while using less than 11 GB memory, and the authors explicitly stated: “We recommend using COMEBin in GPU mode.” These results are consistent with our own measurements, supporting that, **when used as designed on a single GPU, COMEBin’s runtime is comparable to other neural-network-based tools and remains well within a practical range for large-scale applications.**

Although the runtime remains substantially longer than the 5 minutes required by MetaBAT2, COMEBin’s ability to recover more than twice as many high-quality genomes (505 vs. 204) justifies the computational cost. Moreover, once a comprehensive reference genome catalog is established, it can serve numerous downstream analyses for many years, making the investment in GPU resources worthwhile. For instance, a genome binning project involving 1,000 metagenomic samples would require approximately 4 days using 10 GPUs or ~33 hours with 30 GPUs. These calculations demonstrate that GPU-accelerated environments enable such analyses to be completed within realistic project timelines. Given the substantial improvement in genome recovery achieved by COMEBin in our benchmark, we believe its computational demand is well justified by its performance. We have added this discussion to the assessment of computational efficiency across binners in the revised manuscript as follows:

(Line 295) *“Improving genome recovery often relies on computationally intensive algorithms that incur high computational costs, posing challenges for large-scale metagenomic projects. To assess computational efficiency, we benchmarked tools on the Gut 7.2 Gb dataset using a 5-core CPU configuration (Methods). COMEBin and binny showed markedly longer runtimes than other tools (Fig. 4e, Supplementary Table 5), likely reflecting COMEBin’s data augmentation and binny’s iterative processing. With GPU-based computing now widely accessible, a GPU-accelerated COMEBin achieves comparable performance to other neural network-based binners (~1 h per sample) and enables practical large-scale analyses (e.g., 1,000 samples in ~4 days using 10 GPUs). Although its runtime remains longer than MetaBAT2 (5 min), COMEBin’s recovery of more than twice as many genomes (505 vs. 204) justifies the computational investment. Once established, such reference genome catalogs support downstream analyses for years, further validating the value of GPU resources.”*

We thank the reviewer for this constructive and insightful comment. To clarify that this study emphasized genome recovery rather than computational efficiency, we have revised the text accordingly in the manuscript.

(Line 26) *“Neural network-based tools consistently outperformed others in genome recovery from both real samples and simulated samples with realistic taxonomic complexity, though at higher computational cost.”*

(Line 260) *“For genome bins with contamination < 5%, COMEBin consistently outperformed other tools in human gut and soil samples (Fig. 4a). Consistent with the simulated data results, neural network-based tools generally showed superior performance, ...”*

→ *“For genome bins with contamination < 5%, COMEBin consistently recover more genomes than other tools in human gut and soil samples (Fig. 4a). Consistent with the simulated data results, neural network-based tools generally showed superior genome recovery, ...”*

(Line 404) *“Based on these findings, we propose an optimal ensemble binning pipeline that combines COMEBin, SemiBin2-multi, and VAMB-multi, followed by bin refinement using MetaWRAP.”*

→ *“Focusing primarily on maximizing genome recovery, we propose an optimal ensemble binning pipeline that combines COMEBin, SemiBin2-multi, and VAMB-multi, followed by bin refinement using MetaWRAP.”*

(Line 496) *“Neural network-based binning tools consistently demonstrated robust performance across varying sequencing depths and taxonomic complexities, ...”*

→ *“Neural network-based binning tools consistently recovered more high-quality MAGs across varying sequencing depths and taxonomic complexities, ...”*

In addition, we have expanded the Discussion section to **address the potential trade-off between binning performance and computational efficiency, noting that lightweight binners such as MetaBAT2 can also be highly useful depending on the project scale and objectives** as follows:

(Line 502) *“Our benchmark study primarily focused on evaluating binning tools in terms of genome recovery performance. However, improved recovery was often achieved through computationally intensive algorithms, such as deep neural network-based learning (e.g., COMEBin, SemiBin2, VAMB). This trade-off between yield and efficiency should be carefully considered when selecting binners for real-world metagenomic projects. Although lightweight binners such as MetaBAT2 generally recover fewer genomes, they remain highly practical for large-scale analyses with limited computational resources.*

The same trade-off between performance and computational cost also applies to the bin refinement process. MetaWRAP yielded more high-quality genomes than DAS Tool or MAGScoT, but required over ten times longer runtime. However, with sufficient CPU cores, bin refinement using MetaWRAP can be completed within a reasonable timeframe. Therefore, **we proposed an optimal ensemble binning pipeline based on MetaWRAP, primarily focusing on maximizing genome recovery.** We have added this discussion to the revised manuscript as follows:

(Line 398) *“We further evaluated computational efficiency of bin refinement tools using a 5-core CPU configuration and found that MetaWRAP required considerably longer runtime (~85 min) than DAS Tool (~8 min) and MAGScoT (~2.5 min) (Supplementary Table 6). Nevertheless, considering that a well-curated genome catalog can serve numerous downstream analyses for years, we believe that this computational investment can be justified when sufficient CPU resources are available (e.g., refinement of 1,000 samples can be completed in 3 days with 100 CPU cores). Focusing primarily on maximizing genome recovery, we propose an optimal ensemble binning pipeline that combines COMEBin, SemiBin2-multi, and VAMB-multi, followed by bin refinement using MetaWRAP.”*

2. Both, this and the recently published benchmark, seem to narrowly focus on binners that yield more high-quality bins, which, in the case of binners that iteratively look for marker genes to refine MAGs, may bias results and neglect lineages discoverable by more unbiased methods. Some of the modern binners appear to be closer to refinement tools than to actual unbiased, marker-gene agnostic binning tools. The authors should be more explicit in recognizing a key scientific gap they found in their analysis: the trade-off between modern binners that take days to yield a percentage increase in MAGs versus highly efficient binners that complete in minutes; so the field of metagenomics can move in the right direction with the future of binning tools, and avoid investing/wasting disproportional amounts of computing resources and energy consumption when using inefficient tools that yield small gains in number of MAGs compared to the computing resource they demand.

We thank the reviewer for raising this thoughtful point. We agree that distinguishing between truly unbiased binning and refinement-like methods is critical for fair evaluation. However, the **top-performing tools in our study do not iteratively adjust or merge bins based on marker genes**, as refinement programs.

Instead, these tools employ marker-gene information only as weak supervision during model training, not as a criterion for post-hoc refinement. Specifically, COMEBin identifies contigs containing single-copy marker genes and treats them as reliable references for guiding semi-supervised representation learning. Marker-gene-labeled contigs are not repeatedly optimized or used to reassign bins, and contigs lacking markers are not excluded from clustering. Thus, COMEBin does not preferentially favor specific taxa or introduce systematic bias. SemiBin2 similarly used marker-gene contigs only to define positive pairs for contrastive learning, and the clustering itself is unsupervised. Also, some programs such as VAMB, do not utilize marker-gene information at all.

Therefore, while some tools (e.g., binny, Metabinner) explicitly perform marker-gene-driven refinement, the top-performing modern binners are better characterized as semi-supervised algorithms that use marker-gene signals as limited guidance during feature learning, rather than as a direct refinement mechanism.

We also acknowledge the reviewer’s concern regarding potential lineage bias. As shown at Fig. 6e, the neural-network-based binners produced not only a larger number of high-quality MAGs but also unique genomic clusters that were not captured by previous methods, indicating the recovery of novel genomic content. Furthermore, our ensemble pipeline expanded phylogenetic breadth, with newly reconstructed genomes evenly distributed across the GTDB-based tree. Importantly, these include less characterized clades such as RF39, TANB77, and UBA1381, suggesting that marker-gene-guided learning does not exclude underrepresented or novel lineages.

We have added a short paragraph to the Discussion acknowledging that, in theory, marker-gene guidance could introduce minor bias, but our empirical results show this effect to be negligible. We also emphasize the reviewer's valuable point about balancing computational efficiency and biological coverage, which we have highlighted in the revised text so that the field can better balance accuracy, cost, and discovery potential in future binning developments.

(Line 509) *“We also note that marker gene–based iterative refinement contributes substantially to the computational overhead observed in some modern binners (e.g., binny, MetaBinner). Furthermore, marker-driven strategies, rather than marker-guided learning, may introduce lineage-specific biases. Therefore, future developments should aim to reduce such dependencies to ensure that performance improvements do not come at the cost of phylogenetic inclusivity.”*

3. The authors identify genomic chimerism as an often overlooked aspect of bin quality and employ the tool GUNC to assess this non-redundant contamination. Their analysis shows that the proportion of chimeric genomes varies between different binning tools, which is an interesting finding that highlights the limitations/variability of relying solely on marker-gene-based assessments. However, I think that caution should be exercised when interpreting GUNC's output. Based on our own thorough testing of this tool, we have found that it can greatly overestimate contamination and the detection of chimeras. Therefore, while the relative differences in chimerism rates across the various binners are an important takeaway from this study, I would be a little more conservative with the wording around the interpretation of GUNC's results.

We thank the reviewer for this valuable comment and agree that GUNC can provide inaccurate estimation of chimerism. A recent study (e.g., Zhou et al., *Microbiome*, 2024, DOI: [10.1186/s40168-024-01848-3](https://doi.org/10.1186/s40168-024-01848-3) Ref. 29) reported comparison of chimerism MAGs to that of single amplified genomes (SAGs), which is expected to have low-level chimerism. This study demonstrated that GUNC and MDMcleaner, an alternative estimator of MAG chimerism, could report contamination levels that differed by up to 43%. Nevertheless, **the same study showed that GUNC default threshold flagged potential chimerism in only ~1% of SAGs but in ~11% of MAGs, suggesting that GUNC retains reasonable discriminative power for detecting contaminating DNA from different taxa.**

We agree that interpretation of GUNC estimates should be approached with caution due to potential overestimation. However, **our study primarily focuses on relative differences in the recovery of chimeric genomes across binning tools, as the reviewer also mentioned, rather than the absolute contamination values.** In response, we have revised the relevant text to moderate our wording and added a new reference, [Ref. 29], to clarify that GUNC values should be interpreted cautiously as follows:

(Line 284) *“These findings underscore the importance of accounting for genome chimerism when evaluating metagenomic binning tools.”*

→ *“These findings suggest that potential genome chimerism should be taken into consideration when evaluating metagenomic binning tools, although the absolute values of contamination rates should be interpreted with caution²⁹.”*

4. The current framing of the results directly compares binning outcomes from single-end versus paired-end sequencing, concluding that paired-end sequencing yields superior results. However, this comparison is less a benchmark of the binning tools themselves and more a reflection of upstream data quality and processing steps, particularly the performance of the contig assembler. The observed reduction in binning efficacy for single-end samples is an expected consequence of poorer contig quality and increased assembly fragmentation, as the assemblers are limited by the lack of spatial linkage information. To maintain the manuscript's focus on the performance of the binning algorithms, I would suggest that the authors reframe this section. Instead of comparing the two sequencing formats directly, the analysis should compare the performance of the different bidders within each format. This approach would answer more relevant questions, such as, "Which binning tools are most effective at recovering genomes from the lower-quality, fragmented assemblies typical of single-end data?" and "Which tools perform best given high-quality paired-end assemblies?" This reframing would provide a more direct and valuable assessment of the bidders' robustness and capabilities under different input conditions.

We appreciate the reviewer's helpful clarification. We agree that the difference between single-end (SE) and paired-end (PE) results largely originates from the upstream assembly step. Our intention was to illustrate how sequencing format influences genome-resolved metagenomics workflows as a whole. We have therefore framed this section to state explicitly that PE sequencing improves contig length and assembly continuity, which indirectly enhances downstream binning accuracy.

To address the reviewer's suggestion, **we generated a new Fig. 6 which contains additionally comparing the relative performance of bidders within SE sequencing data. Across SE assemblies, where contigs were shorter and more fragmented, neural-network-based bidders still outperformed others, while CONCOCT and single-sample SemiBin2 showed competitive performance.** These additions provide a clearer view of each tool's robustness under different sequencing read formats. The **updated comparison is now visualized in Fig. 6d-e**, which summarizes the performance of each bidder for single-end sequencing data and discussed in the revised manuscript as follows:

(Line 365) *"We next evaluated the performance of the same binning tools on single-end sequencing data using 40 human gut metagenome samples with sequencing depths of approximately 3 Gb and 7 Gb (20 samples per depth; **Supplementary Table 3**). Notably, COMEBin, SemiBin2-multi, and VAMB-multi consistently achieved top-tier genome recovery in single-end datasets, mirroring their performance with paired-end sequencing data (**Fig. 6d**). Overall performance metrics showed similar trends, with CONCOCT and single-sample SemiBin2 also exhibiting competitive results (**Fig. 6e**). These findings indicate that COMEBin, SemiBin2-multi, and VAMB-multi represent robust choices for metagenomic binning across both single- and paired-end sequencing datasets at varying sequencing depths."*

5. The manuscript contains a factual error, stating CheckM2 relies on single-copy marker genes, which was true for CheckM1 but not the machine-learning-based CheckM2.

We thank the reviewer for catching this factual error. We have corrected the description to specify that CheckM2 employs a machine-learning model trained on amino-acid composition and genome-feature patterns, rather than explicit single-copy marker approach used by CheckM, as follows:

(Line 257) *“As ground truth genome assignments are unavailable for these real datasets, we evaluated genome bin completeness and contamination using CheckM2, which estimates these metrics through a machine-learning model trained on genome-wide amino-acid composition and gene-content features²⁶.”*

(Line 274) *“Importantly, non-redundant contamination cannot be detected by CheckM2, which relies on single-copy marker genes to estimate contamination.”*

→ *“Importantly, non-redundant contamination is difficult to detect by CheckM2.”*

6. The usage of AI tools for writing is evident in the style and grammar used. I am not against it at all, but the tone of the paper starts to sound very homogeneous and not building excitement

as the manuscript progresses. The authors should at least acknowledge and disclose their use of AI tools, and the specific models used in this case.

According to Springer Nature's policy, authors do not need to declare the use of AI tools when they are used for copy editing. However, we agree that such disclosure improves the transparency of the work. Therefore, as suggested, we have added the following statement in the Methods section.

(Line 639) "***Use of artificial intelligence tools***

The authors used OpenAI ChatGPT-4 to assist with language editing and readability improvement. The authors are solely responsible for the scientific content."

All in all, I consider this manuscript to be of high relevance, and I will be more than happy to review it again provided the authors revise it accordingly. The paper is a valuable contribution that I believe will help guide the design of future metagenomic studies.

We sincerely thank the reviewer for the constructive and encouraging feedback. We have carefully revised the manuscript according to all comments and suggestions, particularly by incorporating an evaluation and discussion of the computational efficiency of the tools and by reframing sections related to the binning of single-end read metagenomic data. We hope that our revisions adequately address the reviewer's concerns.

Reviewer #3 (Remarks to the Author):

This paper benchmarks binning algorithms for reconstructing bacterial genomes from metagenomic datasets, using commonly used datasets, newly simulated datasets, and real metagenomic datasets. The authors report several notable findings, including that newer neural network–based methods outperform earlier approaches. They also propose a new pipeline that integrates and refines genome bins from the top three tools. This pipeline recovered over 30% more high-quality genomes than previous methods.

The paper offers practical guidance for improving metagenomic binning techniques. However, some findings are expected and not particularly novel—for example, that sequencing depth and taxonomic complexity strongly influence performance.

We agree that researchers may expect sequencing depth and taxonomic complexity to influence binning efficacy. However, surprisingly, no previous benchmarking study has systematically examined how these factors affect the performance of different binning tools. **To the best of our knowledge, this is the first benchmarking study that explicitly evaluates multiple key factors—sequencing depth, taxonomic complexity, genome chimerism, the number of samples in multi-sample binning, and sequence-read layout—and their combined impact on binning performance.**

We found that **although sequencing depth generally affects binning efficacy, the degree of its influence varies across binning tools.** Its influence was relatively smaller for some tools, such as COMEBin and VAMB-multi, compared to others, particularly CONCOCT, which showed a marked decrease in performance in samples with high taxonomic complexity. These results provide practical guidance for researchers selecting binning tools for projects with varying levels of microbial diversity.

In addition, **our study represents the first benchmark that considers genome chimerism.** We found that although certain binners retrieved a large number of near-complete MAGs, many of these were actually chimeric genomes. For example, MaxBin2 and MetaBinner produced substantially higher proportions of chimeric MAGs than other tools, whereas VAMB showed only a small proportion.

In summary, our study identifies factors that influence metagenomic binning performance and demonstrates how the magnitude of their impact varies across tools, providing essential guidance for selecting the most suitable binning approach for a given dataset or research objective.

More detail is needed on the newly simulated datasets. Simply stating that "CAMISIM24 (v1.3.0) with a de novo community design" and "default parameters was used" is insufficient. Information about the microbial community complexity (e.g., gut, oral, or other environments), taxonomic composition, and abundance distributions represented in the simulated datasets would be valuable.

We appreciate the reviewer's helpful suggestion. We fully understand the request for additional details about the composition and complexity of the simulated datasets. However, **our simulated datasets were not designed to replicate any particular microbial community (such as gut, oral, or skin microbiome), but rather to serve as general-purpose synthetic communities** for method benchmarking. In other words, we intentionally generated artificial

metagenomes from a diverse collection of reference genomes to provide a neutral, controlled basis for tool comparison, rather than modeling a specific ecological niche. **This design choice and its limitations are already described in the initially submitted manuscript.** In particular, the Limitations section explicitly notes that the simulated datasets were not constructed to represent any real microbial community, as follows:

(Line 530) *“This study has several limitations. First, the simulation datasets were designed to achieve high taxonomic complexity by incorporating NCBI bacterial genomes from diverse environments. While this approach ensures broad diversity, it does not specifically mimic any single environmental condition, which may limit its direct applicability to certain real-world ecosystems. However, the inclusion of real environmental datasets and the CAMI human dataset, which yielded consistent results, helps mitigate this limitation.”*

In response to the reviewer’s suggestion, **we added new supplementary figures illustrating the phylum-level taxonomic composition across all simulated datasets (Supplementary Fig. 1a)** and one of their corresponding abundance distributions, which follow a log-normal pattern (**Supplementary Fig. 1b**). These additions improve the transparency of our dataset design and provide a clearer view of the diversity represented in our synthetic communities.

Certain aspects of the benchmarking could also be strengthened. For example, the paper concludes that multi-sample binning is most effective with around 20 samples, as too few or too many samples reduce its benefits. However, this conclusion appears to be based on a single collection of simulated datasets. It remains unclear whether this result would hold under different microbial community complexities or sequencing depths.

We thank the reviewer for this constructive comment. To assess whether the observed optimum around 20 samples is consistent across datasets with different taxonomic complexities and sequencing depths, **we repeated the evaluation using the Gut 4.5 Gb dataset, which contains samples with a different sequencing depth** from the first test dataset (Gut 7.2 Gb,

Fig. 5a–b) and likely varying taxonomic complexities. From this additional analysis, we observed that both VAMB-multi and SemiBin2-multi exhibited the same saturation trend around 20 samples, beyond which performance plateaued or slightly decreased. These new results have been added to the revised manuscript as **Fig. 5c–d** along with related text as follows:

(Line 323) *“Repeating the evaluation with the Gut 4.5 Gb dataset produced consistent trends (Fig. 5c-d).”*

Additionally, similar pattern in multi-sample binning performance was reported in recent Vaginal catalog study (<https://www.nature.com/articles/s41564-024-01751-5>), which is based on different microbial community (vagina) with generally lower taxonomic complexity than gut. These results independently support the idea that the 20-sample optimum reflects a general feature of current multi-sample learning strategies rather than a dataset-specific artifact.

Thank you again for your time and effort, and for assisting in improving the manuscript. We look forward to your positive response.

Responses to referees (our responses in bold or colored text)

Manuscript ID: NCOMMS-25-34019A

Attached please find our revised manuscript. We do appreciate the reviewers' critical but valuable comments, which have been a guideline in revising the manuscript. We have provided a revised manuscript. We believe that the revisions improved the manuscript particularly in delivering our analysis to the readers clearly. We hope that our revised manuscript is suitable for the publication.

=====

REVIEWER COMMENTS

Reviewer #2 (Remarks to the Author):

1. I do not agree that COMEBin (or MetaBinner) should be benchmarked together with real binning tools. COMEBin uses CheckM marker genes to build/cluster bins, COMEBin pre-filters binning results using CheckM, and then the benchmarking is done using CheckM... it is just not an independent or fair way to compare binning performance, because the method is effectively being evaluated with the same marker framework it is built around.

We thank the reviewer for raising an important concern regarding benchmarking independence when single-copy marker gene (SCG)-based criteria are used both for result selection and for performance evaluation.

First, we would like to clarify that COMEBin does not use CheckM marker genes as features for contig representation learning or as the primary basis for clustering. As described in the Methods of the COMEBin study, contigs are clustered based on k-mer composition and coverage profiles, while SCGs are only used as structural constraints to prevent contigs harboring the same SCG from being merged during Leiden clustering.

We agree with the reviewer that COMEBin (and MetaBinner) employ SCG-based quality estimates to select a final binning result among multiple candidates, and that benchmarking such methods using the marker-gene-informed evaluation (e.g., CheckM2) raises a valid concern regarding evaluation independence.

Importantly, the concern is not the use of SCG-based quality assessment itself, but the potential overlap between optimization criteria within a method and the metrics used for benchmarking.

In this study, our benchmarking goal was not limited to comparing intrinsic clustering ability in isolation, but rather to reflect realistic genome-resolved metagenomic workflows. In practice, real datasets lack ground truth, and SCG-based quality control is often the only feasible means to assess bin quality. Consequently, quality-aware workflows, where bins are generated, evaluated, refined, or selected based on SCG-based criteria, are commonly used by practitioners.

We note that many widely used binning tools utilize SCGs in different ways, including (i) using SCG-containing contigs as initial seeds or anchors (e.g., MetaDecoder, MaxBin2), (ii) performing bin-level validity checks or refinement based on SCG duplication (e.g., binny, SemiBin2), and (iii) selecting a final result among multiple candidates based on SCG-derived quality scores (e.g., COMEBin, MetaBinner). While these strategies differ in scope, all introduce some degree of dependence on SCG-based signals.

P
A
G
E

Crucially, we further demonstrate using simulated datasets that COMEBin achieves strong performance under ground-truth-based evaluation (AMBER), which is independent of SCG-based quality metrics. This orthogonal assessment unaffected by SCG-driven model selection indicates that the performance of COMEBin cannot be attributed solely to marker-gene-based result selection.

Nevertheless, we acknowledge that the use of SCG-based result selection represents a limitation from a strict benchmarking-independence perspective. **We have therefore explicitly discussed this point in the Discussion section as a limitation of SCG-aware binning strategies as follows.**

(Line 539) *“Third, some tools (e.g., COMEBin and MetaBinner) use single-copy marker genes (SCGs) to select a final binning output among multiple candidates, which may partially overlap with marker-gene-informed evaluation on real datasets (e.g., CheckM2) and complicate strict benchmarking independence. Although we acknowledge this potential concern, we mitigate it by additionally evaluating performance on simulated datasets using ground-truth-based metrics (AMBER), which are independent of marker-gene-based quality estimates and therefore provide an orthogonal assessment unaffected by SCG-driven model selection.”*

Overall, while we agree that SCG-based result selection complicates strict independence in benchmarking, we believe that including such methods is necessary to provide a comprehensive and user-relevant comparison of state-of-the-art binning pipelines.

2. The authors addressed all my comments.

We thank the reviewer for time and effort in reviewing the manuscript.

Reviewer #3 (Remarks to the Author):

The authors have fully addressed my concerns in this revision.

We thank the reviewer for time and effort in reviewing the manuscript.

P
A
G
E